# TraffiDent: A Dataset for Understanding the Interplay Between Traffic Dynamics and Incidents

**Xiaochuan Gou**[1][*] **Ziyue Li**[2][*] **Tian Lan**[3], **Junpeng Lin**[3], **Zhishuai Li**[4],
**Bingyu Zhao**[5], **Chen Zhang**[3][†] **Di Wang**[1], **Xiangliang Zhang**[1,6][†]

[1]King Abdullah University of Science and Technology, Saudi Arabia
[2]Technical University of Munich, Germany
[3]Tsinghua University, China
[4]Institute of Automation, Chinese Academy of Sciences, China
[5]Vienna University of Technology, Austria
[6]University of Notre Dame, USA

## Abstract

Long-separated research has been conducted on two highly correlated tracks: traffic and incidents. Traffic track witnesses complicating deep learning models, e.g., to push the prediction a few percent more accurate, and the incident track only studies the incidents alone, e.g., to infer the incident risk. We, for the first time, spatiotemporally aligned the two tracks in a large-scale region (16,972 traffic nodes) from year 2022 to 2024: our TraffiDent dataset includes **traffic**, i.e., time-series indexes on traffic flow, lane occupancy, and average vehicle speed, and **incident**, whose records are spatiotemporally aligned with traffic data, with seven different incident classes. Additionally, each node includes detailed physical and policy-level meta-attributes of lanes. Previous datasets typically contain only traffic or incident data in isolation, limiting research to general forecasting tasks. TraffiDent integrates both, enabling detailed analysis of traffic-incident interactions and causal relationships. To demonstrate its broad applicability, we design: (1) post-incident traffic forecasting to quantify the impact of different incidents on traffic indexes; (2) incident classification using traffic indexes to determine the incidents types for precautions measures; (3) global causal analysis among the traffic indexes, meta-attributes, and incidents to give high-level guidance of the interrelations of various factors; (4) local causal analysis within road nodes to examine how different incidents affect the road segments' relations. The dataset is available at https://xaitraffic.github.io.

## 1 Introduction

In today's era of deep learning, a technological foundation has been laid for intelligent transportation systems [55, 58, 30]. Primarily, conducting myriad traffic analysis relies on two types of data: traffic and incident data. Traffic data encompasses the traffic state-related time-series, e.g., volume, speed, and occupancy rate on the road network over time. This continuous stream of data is essential for forecasting the future volume, understanding peak usage times, and optimizing traffic signals and routes [16, 25]. Real-time traffic data allows for dynamic adjustments to be made, enhancing the efficiency of traffic flow and reducing overall travel times. On the other hand, incident data includes information about traffic accidents, road closures, and unexpected events that can significantly affect traffic flow. This data helps in understanding the impact of such incidents on traffic congestion

---

[*]Equal contribution.
[†]Corresponding authors.

39th Conference on Neural Information Processing Systems (NeurIPS 2025) Track on Datasets and Benchmarks.

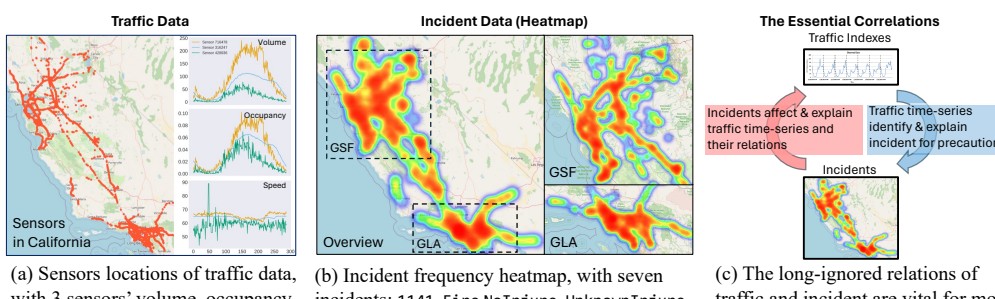

| Traffic Data | Incident Data (Heatmap) | The Essential Correlations |
|---|---|---|

(a) Sensors locations of traffic data, with 3 sensors' volume, occupancy rate, and speed indexes

(b) Incident frequency heatmap, with seven incidents: `1141`, `Fire`,`NoInjure`, `UnknownInjure`, `Hazard`, `AnimalHazard`, `CarFire`

(c) The long-ignored relations of traffic and incident are vital for more explainable and critical analysis

Figure 1: Our TraffiDent contains (a) traffic data with road-level meta features, (b) incident data, and (c) their essential yet neglected relations

and travel time, facilitating more accurate predictions and enabling timely responses from traffic management systems [26, 29]. By analyzing patterns and frequencies of incidents, predictive models can also be developed to foresee potential hotspots and prevent future occurrences.

However, current research has been conducting the two tracks of traffic and incident separately, ignoring the inseparable relation of traffic and incident. For example, abundant works [44, 21, 12, 60] have been using various **traffic-only** datasets such as PEMS [46], META-LA [46], LargeST [32] for traffic forecasting. They achieved relatively high accuracy because, under normal circumstances, traffic flow generally follows a strong regular temporal pattern. However, they ignore that unexpected incidents will cause abnormal and irregular patterns in traffic flows. FT-AED [8] provides anomaly annotations but is limited in scale, covering only 196 sensors. Moreover, it lacks detailed incident labels and location features, making it suitable only for coarse-grained time series anomaly detection tasks without support for detailed incident analysis. On the contrary, in the **incident-only** data [2, 36], studies have been done on offering descriptive analysis on the incident patterns [24, 1], predicting the accident risk [45], time-to-accident [3], or next incident [19], yet **the biggest research gap** is: there is very limited work supporting the research of interrelations between the traffic time-series and incidents, especially to identify and explain the incidents and their causal relations with the traffic systems and roads. Moreover, the data granularity of existing open-source datasets [10, 38, 39, 19] is extremely coarse, and there is no specific location information, such as coordinates or absolute postmile (Abs PM) markers. These factors make conducting related research particularly challenging. Some traffic studies that incorporate incident data use datasets that have not been aggregated or made open source, making it difficult to use them as a standard for evaluating new methods. Additionally, due to the issue of large granularity, it's impossible to analyze the specific impact of accidents on precise road segments. Instead, incident data can only be used to predict general volume within a certain area.

**Contributions**. To address the research gaps, we introduce the TraffiDent dataset. This dataset not only includes three distinct types of traffic time series data for the entire 3 years from 2022 to 2024 (in Fig. 1(a)), but also encompasses comprehensive incident data (in Fig. 1(b)) and meta-features of roads closely related to traffic flow. The contributions of this dataset can be summarized as follows: **(1) We provide a comprehensive collection of multi-type incident records with 1,441,904 samples.** It enables the training and evaluation of traffic forecasting models across various scenarios/incidents. This also supports tasks such as incident discovery and traffic anomaly detection by providing ground truth data. **(2) We offer a rich collection of physical and policy-level road meta-features.** These features are instrumental for causal analysis of traffic and support the increasingly popular field of interpretable deep learning models. **(3) By incorporating rich, fine-grained attributes, our dataset enables researchers to explore the underlying mechanisms driving traffic dynamics and model behavior.** Furthermore, we introduce four novel tasks and corresponding experiments that are infeasible to implement or evaluate using existing datasets. As shown in Fig. 1(c), our TraffiDent helps not only **Incident → Traffic**: e.g., to analyze how incidents affect the traffic states (with our post-incident traffic forecasting in Sec. 4.2) and traffic node relations (local causal analysis in Sec. 4.5), but also **Traffic → Incident**: e.g., to identify the incidents (incident classification in Sec. 4.3) and explain the incident with other factors from the system (global causal analysis in Sec. 4.4).

To our knowledge, TraffiDent is the most recent large-scale dataset that simultaneously includes traffic, incident, and meta-feature information in terms of collection time, covering three distinct types of traffic volume. This ensures the timeliness of traffic research, providing a robust foundation

for studies aiming to capture and explain traffic dynamics, causation, and interrelations. TraffiDent serves as a rigid testing bed and empirical support to justify model effectiveness and interoperability in deep learning and the traffic community.

## 2 Related Work

### 2.1 Related Work of Traffic and Incident Datasets

**Traffic Dataset**. Traffic dataset are commonly used in traffic analysis and forecasting as experimental benchmarks. We introduce the existing four public datasets widely leveraged in traffic forecasting experiments. The PeMSD7(M) and PeMSD7(L) are proposed by [55]. METR-LA and PEMS-BAY[27] covered similar regions in California with multiple traffic. However, these datasets are limited to one collection region, our TraffiDent instead covers the majority of Metropolitan regions of California including greater San Francisco, San Jose, and Greater Los Angeles. The PEMS03,04,07 and 08 are proposed by [46]. This dataset encompasses four different regions, with data collected using the same rules, supporting multi-region research for traffic analysis and forecasting. Compared to previous datasets, LargeST [32] extends both temporally and spatially and includes some basic meta-features. However, these datasets still lack important features such as incidents that significantly impact traffic, as well as road meta-features related to physics and policy. Our TraffiDent is the first dataset as such that spatiotemporally align traffic dynamics and traffic incidents, enabling uncomparable potentials for explainable and interpretable traffic management tasks.

**Incident Dataset**. Incident datasets support the traffic analysis, like the incident impact on traffic, and incident detection. [19] proposes a dataset that includes accident data with accident relative features, like accident reason. [54] also leverages a large accident dataset including various types for accident hotspot prediction. However, limited to the absence of traffic time series, it is hard to do a deeper impact analysis of accidents on traffic based on these datasets. [29] leverages a dataset that includes 13,338 accident records with the traffic flow. However, the dataset is small and non-public. [60] proposes a new accident prediction model based on a dataset with accidents and traffic flow. However, the dataset is not public, either.

### 2.2 Traffic and Incident Analysis

**Traffic Forecasting with Incidents Considered**. A large number of works, e.g., STGCN [55], STGODE [7], DSTAGNN[21], are proposed to improve the prediction accuracy based on GNN [52] and RNN [41] models. However, these works only consider historical traffic for future traffic, yet other critical impacts, e.g., incidents and meta-features are ignored ([56, 20, 47, 32] offer detailed reviews in traffic forecasting). There are a few works that have considered incidents when predicting, whose main design is incorporating incident-related embedding as auxiliary information into traditional spatiotemporal prediction framework [53, 14, 33, 18]. For example, DIGC-Net [53] inputs the type and duration of the incident to predict the affected speed. Yet, the dataset only brings one week of incident data (17-24 Apr 2019) from a small district, being spatiotemporal limited; STCL [33] introduced two-month New York City *Vehicle* incident data as one-hot accident embedding into the prediction of the *Taxi* and *Bike* data. Like what we have observed in most works that analyze traffic with incidents [33, 18], **the transport modes of traffic data and that of incident data are NOT seamlessly matched**; thus, it will be less convincing to analyze vehicle incidents' impact on bike traffic (bike lane is separated from vehicle lane) or on taxi traffic (taxi is only a subset mode of the whole vehicle). Our TraffiDent is the first and only dataset that is (1) spatially and temporally large-scale, (2) that modes in traffic and incident are seamlessly matched (all vehicles), which guarantees unbiased analysis between the traffic and incident.

**Incident Classification**. It is a crucial task in analyzing non-recurrent congestion [26]. Recently, several studies have utilized traffic flow data for incident classification [29, 60]. However, these studies often face at least one of the following three challenges: (1) Small Data Size [29]: Many studies suffer from limited datasets that fail to capture the diversity and complexity of traffic patterns, affecting model reliability and generalizability. (2) Limited Dimensionality without Traffic Volume Data [19, 54]: The absence of critical data dimensions, such as traffic volume, restricts the depth and accuracy of incident classification models. (3) Experiments Based on Non-Public Dataset [60, 29]: Reliance on proprietary datasets impedes the ability of the broader research community to verify, replicate, or enhance the findings, limiting collaborative advancements.

**Traffic Causal Analysis**. This task aims to learn the causal structures among different entities in a traffic system. Usually, the causal structures are formulated as Bayesian networks or DAGs [59],

where a directed edge denotes the causal link. In the traffic domain, given the traffic indexes are time-series and others can be scalers (e.g., static meta-features), a special DAG structure learning based on heterogeneous data is needed [22, 23]. **To learn the global relation of various factors**, e.g., traffic flow, meta-attributes, weather, etc., where each DAG node is a factor to be considered, MultiFun-DAG [23] views multivariate time-series in traffic as a multi-function and formulate the structure learning as a "self-expression problem", i.e., $\mathbf{X} = \mathbf{WX} + \mathbf{Z}$, based on function-to-function regression and the directed acyclic regularization on the coefficients $\mathbf{W}$ [59], a DAG is constructed based on $\mathbf{W}$. MM-DAG [22] further consider multi-location at the same time. **To learn local causal relation among different locations**, where each DAG node is a spacial location, DBGCN [34] and DCGCN [28] combines DAG with GCN, DCGCN further considers the causal links across the time, i.e., node 1 at $t_1$ affects the node 2 at $t_3$, and the dynamics of DAG changing over time. The preliminary of these four intended tasks are introduced in Appendix A.4.

## 3 TraffiDent Datacube

### 3.1 Comparison with Existing Datasets

Table 1: The comparison of Existing Traffic Dataset and TraffiDent. Each row represents the largest subset within the corresponding dataset.

| Dataset | Nodes | Edges | Slot (Min) | Location | Context | Physics | Policy | Gran. | Incdt. |
|---|---|---|---|---|---|---|---|---|---|
| PeMSD7(L) | 1,026 | 14,534 | 5 | ✓ | ✓ | - | - | Road | |
| METR-LA | 207 | 1,515 | 5 | ✓ | - | - | - | Road | - |
| PEMS-BAY | 325 | 2,369 | 5 | ✓ | - | - | - | Road | - |
| PEMS07 | 883 | 865 | 5 | - | - | - | - | Road | - |
| CA | 8,600 | 201,363 | 15 | ✓ | ✓ | - | - | Road | - |
| TraffiDent | 16,972 | 870,100 | 5 | ✓ | ✓ | ✓ | ✓ | Lane | ✓ |

Table 2: The comparison of Incidents Dataset and TraffiDent. We are the only ones who combine traffic with incidents.

| Dataset | Incident | Gran. | Volume | Speed | Occupancy |
|---|---|---|---|---|---|
| CTC [10] | 1 | Point | - | - | - |
| NYC Col [38] | 1 | Road | - | - | - |
| NYS Crashes [11] | 1 | Point | - | - | - |
| UKA [39] | 1 | Point | - | - | - |
| TAP [19] | 1 | Road | - | - | - |
| TAA [5] | 1 | Road | - | - | - |
| TraffiDent | 7 | Point | ✓ | ✓ | ✓ |

(1) **Base Features Comparison**. In Table 1, we introduce the existing four public datasets widely leveraged in **traffic analysis**. The PeMSD7(M) and PeMSD7(L) are proposed by [55]. The PEMS03,04,07 and 08 are proposed by [46]. The large-scale traffic dataset LargeST which includes CA, GLA, GBA, and SD subdatasets are proposed by [32]. We compared the datasets from 7 aspects: Scale (Number of Sensors/Nodes and Neighbors/Edges), Location (Latitude, Longitude, Abs PM), Context (Road Name, City, County), Physics Meta Feature (Road Width, Terrain, Surface Material, etc.), Policy meta feature (Design Speed Limit, Population, Functional Class etc.), Granularity (timer interval, sensor level) and Incident features. As shown in Table 1, our TraffiDent is larger than the existing datasets, with 16,972 nodes and 870,100 edges. Compared to other datasets, we include two types of meta features: physics meta feature and policy meta feature. The physics meta feature details the tangible, structural characteristics of a road, and the policy meta feature is fundamental for the operational and planning purposes of a road. These features provide strong support for constructing interpretable traffic forecasting and traffic causal analysis.

In Table 2, there are 6 famous existing datasets for **incident analysis**. Compared with them, our TraffiDent includes 7 incident categories besides accidents. The category feature provides fundamental support for studying the impact of different incidents on traffic and also offers ground truth for detecting incidents beyond accidents. Also, TraffiDent includes three kinds of traffic time series: traffic volume, road occupancy rate, and vehicle average speed. Such an integration significantly broadens the scope of analyzing post-incident impacts.

(2) **Comprehensive Road Meta Features in Multiple Aspects**. As shown in Table 1, TraffiDent dataset includes a wide range of road meta-features, categorized into context, location, policy, and physics-related features. Context features include attributes like district and county, which help understand traffic patterns within different administrative or urban contexts. Location features provide precise spatial attributes such as road coordinates and segment information. Policy features cover regulatory aspects like speed limits, while physics features address road characteristics such as terrain type. This diverse set of meta-features allows for a holistic analysis of how various factors influence traffic behavior.

(3) **The Bridge between Traffic and Incidents**. TraffiDent bridges the gap between traffic and incident data by integrating detailed traffic flow metrics (such as lane-level flow, speed, and occupancy) with comprehensive incident records (including various types of accidents). **From Traffic View**. As shown in Table 1, the existing datasets often lack incident data and have insufficient road-level granularity to effectively study the impact of incidents on traffic. For example, the effect of a traffic

incident on one side of the road might be significantly different from the impact on the other side. TraffiDent offers lane-level traffic flow, speed, and occupancy data with incident features, allowing for detailed analysis of traffic patterns under incident impact at a micro level. Also, the high resolution supports the identification of specific traffic bottlenecks, congestion patterns, and variations in traffic behavior that aggregate or road-level datasets might obscure. Researchers can perform fine-grained analysis to understand traffic dynamics under different incident impacts with greater precision. **From Incident View**. Compared to the existing opensource incidents dataset, as shown in Table 2, TraffiDent has two advantages: (a)Not like other datasets only include accident type incidents, TraffiDent covers 7 specific incident types, such as hazzards and road closures. (b)TraffiDent includes three traffic time series (flow, speed, occupancy) more than other public datasets, even more than some non-public datasets [29, 60]. This allows for sophisticated analyses of how various incidents impact traffic conditions across different regions and times. Researchers can study patterns like the frequency and severity of incidents, their spatial distribution, and their temporal effects on traffic flow, providing a nuanced understanding of incident impacts that single-dataset approaches might miss. By combining incident data with detailed traffic metrics, our dataset facilitates advanced causal analysis, enabling researchers to determine how specific incidents influence traffic behavior over time. This capability supports the development of more effective traffic management strategies and predictive models that account for the immediate and long-term effects of incidents on traffic conditions.

### 3.2 Collection and Construction

Both incident and traffic data are collected from Caltrans Performance Measurement System (PEMS). We started our collection on April 20, 2024, and ended on May 10, 2024. The time span of the data covers the entire 3 years from 2022 to 2024. For **traffic data**, we removed the sensor with less than $50\%$ observations of traffic volume and reserved the data of 16,972 sensors with meta-features. These sensors are located in 42 different cities and counties. We also collected comprehensive meta-features of these sensors. After excluding the features with the same value and features unrelated to traffic, 27 meta-features are reserved. These meta-features can be divided into 5 types as shown in Table 1. Full meta-features are in Table 6, Appx. A.3. As most methods in traffic forecasting are graph-based, the adjacency matrix is a key component for the model to learn spatial dependency. The construction of adjacency matrix is introduced in Appendix A.6.

For **incident data**, we removed repeated incident records and the records without absolute postmile (indicating the position and date-time). As the source and CA PM (we have Abs PM to locate the incident) are relatively redundant in the traffic analysis, thus also being removed. The reserved incident data includes 1,441,904 samples with 9 features. Identifying which nodes are impacted by an incident is crucial for leveraging incident records in traffic analysis. To facilitate this, we use a method that combines the freeway name and absolute postmile (Abs PM) markers to pinpoint sensors that might be affected by the incident. We provide two methods for this matching process: (1) involves matching only the nearest sensor on the same freeway as the incident, (2) involves setting a distance threshold and incorporating all sensors within this specified range.

**Data imputation** is also a crucial aspect of dataset application. However, Considering that different researchers may prefer different data cleaning methods (for example, to handle missing data, some may prefer zero filling for tasks such as classification, while some may avoid zero filling and prefer linear interpolation if the task is imputation so that they don't mix the ground-truth zero value and missing value). Thus, we believe providing the raw data we collected is better. In the experiments mentioned in Sections 4.2 and 4.3, we used zero-filling to address missing values in the traffic series features. More data imputation methods applications are discussed in Appendix A.5.

## 4 Experiments

### 4.1 Descriptive Analysis of TraffiDent

To demonstrate the correlations between traffic conditions and incidents across various spatial and temporal dimensions, we conducted separate analyses for both traffic data and incident data in Year 2023. The cross year data analysis can be found at https://xaitraffic.github.io.

#### 4.1.1 Traffic Trending and Peak Hour Analysis

**Traffic Time Series Variation Patterns**. We defined weekdays as Monday through Friday and weekends as Saturday and Sunday. Within each group, we averaged traffic data from all sensors for the same time interval and normalized the data within each traffic category for visualization. As shown in Fig. 2(a) and (b), the results reveal a distinct evening peak in traffic between 4:00-5:00

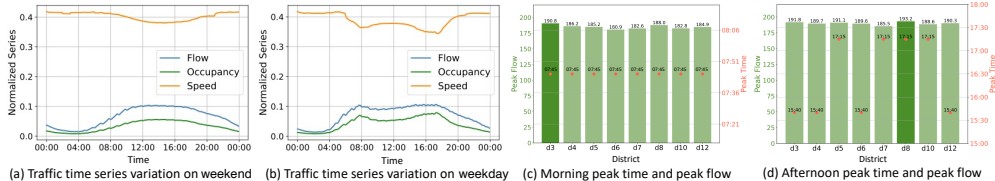

Figure 2: (a) and (b) represent the average traffic variation of all sensors on weekends and weekdays, respectively. (c) and (d) represent the average peak flow and the peak time in different districts. The peak flow is calculated based on all sensors in the specific district. Deeper green indicates the highest peak flow.

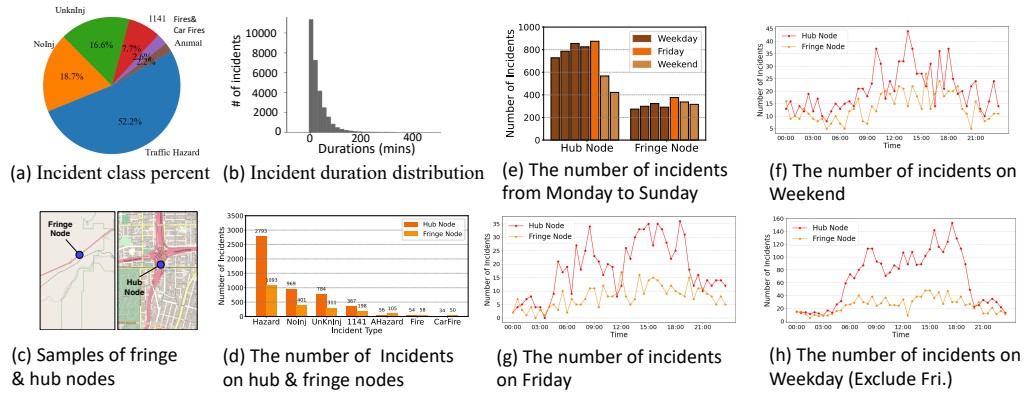

Figure 3: Descriptive analysis of Incidents.(a) and (b) are calculated based on all incident records excluding Type Other. (c)-(h) are calculated based on the incidents happening on hub/fringe nodes.

PM on weekends. In contrast, weekdays exhibit both morning and evening peaks, with high traffic levels sustained throughout the day. Additionally, the statistical analysis shows an inverse relationship between speed and occupancy rate, as well as flow.

**Traffic Peak Hour Analysis**. We define the morning peak hour as between 6:00 and 10:00 and the evening peak hour as between 15:00 and 20:00. We calculated the average flow for each time interval across all sensors in different districts during peak hours and identified the peak flow values along with their corresponding peak times. As shown in Fig. 2(c) and (d), the peak times in the morning are consistent across districts, with similar traffic flow patterns. In the afternoon, however, two distinct peak times are observed: 15:40 and 17:15. Since our flow data is recorded by time intervals, 15:40 and 17:15 represent the time intervals 15:40-15:45 and 17:15-17:20, respectively.

### 4.1.2 Incidents Analysis on Hub and Fringe Nodes

At first, we summarize the distribution of incident durations and types. Fig. 2(a) reveals a long-tail distribution where most incidents are relatively short, but a few incidents last for an extended period. It also demonstrates the geographical distribution of incidents, with higher concentrations in urban areas. The pie chart in Fig. 3(b) shows hazards constitute the majority (52.2%). Next, we aim to further explore the frequency of incidents occurring on road segments under different levels of congestion. In the road network, each sensor can be considered as a node on a specific road. The hub node represents busy intersections and main roads, while the fringe node represents roads in remote areas or branch roads. Two cases for hub node and fringe node are shown in Fig. 3(c). Then, we conduct the following two analyses to reveal the relationship between different types of roads and incidents. (Details in Appendix A.7).

**The Frequency of Different Types of Incidents Next to Hub and Fringe Nodes**. We tally the total number of various types of incidents occurring next to hub nodes and fringe nodes. Fig. 3(d) shows that hazards and accidents are significantly more frequent on busy roads. In contrast, incidents related to fires and hazards caused by animals are more common on remote and less-traveled roads. This may be attributed to the higher activity of animals in less frequented areas and the increased risk of fire in remote areas due to reduced human attention. Based on the observations above, we find that

the incident patterns for hub nodes and fringe nodes differ significantly. We are now curious about whether there are also differences in incidents between these two types of nodes in terms of time.

**Incident Temporal Patterns on Hub and Fringe Nodes**. Firstly, we analyze the distribution of incidents for two types of nodes throughout the week. Without considering incident types, we aggregated incidents by the day of the week. As shown in Fig. 3(e), there is a significant difference between hub nodes and fringe nodes. For **hub nodes**, the proportion of incidents on weekdays is higher than on weekends. This is expected, as downtown areas experience greater traffic volumes and are more prone to accidents on weekdays due to the high traffic flow in main roads and densely populated areas. On weekends, many people move to rural areas, reducing traffic pressure and leading to fewer incidents. In contrast, for **fringe nodes**, we can observe a reversed trend: there is a slight increase in the number of incidents on weekends compared to weekdays: this might be because the residents are moving back to rural areas for the weekends, thus bringing higher possibility of incidents. Interestingly, both types of nodes experience a peak in incidents on Fridays. Such a "Friday mood" will universally increase the incident risk regardless hub nodes or fringe nodes. Moreover, we conducted further analysis on the 30-minute variation of incidents. To analyze the differences in incident patterns between weekdays, Fridays (the day with the highest number of incidents), and weekends, we examined the variation in incident numbers throughout the day for both hub nodes and fringe nodes. The results, shown in Fig. 3(f), (g), and (h), reveal that the number of incidents on hub nodes and fringe nodes varies significantly on weekdays. Incidents typically occur during morning and evening peak hours. Compared to other weekdays, incidents on Fridays show greater fluctuations, likely due to Friday being a transitional day between weekdays and weekends.

## 4.2   Traffic Forecasting after Incidents

The existing models are effective in general traffic forecasting tasks. However, their performance under irregular volumes caused by incidents has not been thoroughly discussed. To assess these models' response in such conditions, we conduct irregular traffic forecasting based on TraffiDent.

**Experiment Setting:** We selected prediction samples from the test set that one incident occurred within a 5-minute window. Due to the large volume of data, we chose to conduct experiments using traffic volume data from the San Bernardino (561 mainline sensors) within the TraffiDent dataset for the first 3 months. All of the baselines are state-of-the-art in the spatial-temporal forecasting or traffic forecasting domain. Our forecasting experiments were implemented within the same software framework employed by [32].

Table 3: The results in different horizons in Monterey (D5 Area). 'General' shows the performance of the model across all samples in the test set, while 'Incident' is on samples after an incident has occurred, with the top 1 in grey, 2nd in boldface, and 3rd underlined.

| Test | Model | 5 Mins (t=1) | | | 15 Mins (t=3) | | | 30 Mins (t=6) | | |
|---|---|---|---|---|---|---|---|---|---|---|
| | | MAE | MAPE | RMSE | MAE | MAPE | RMSE | MAE | MAPE | RMSE |
| General | LSTM | 12.58 | 11.81 | 21.45 | 15.41 | 14.21 | 26.29 | 18.68 | 18.03 | 31.54 |
| | ASTGCN | 12.45 | 13.11 | 20.90 | 14.59 | 13.66 | 23.10 | 16.03 | 15.56 | 27.82 |
| | DCRNN | 11.90 | 11.82 | 20.47 | 13.41 | 12.92 | 23.79 | 14.84 | 14.32 | 26.74 |
| | AGCRN | 12.54 | 12.56 | 22.65 | 13.55 | 13.18 | 25.27 | 14.62 | 14.24 | 27.92 |
| | GWNET | 11.99 | 11.85 | 20.30 | 13.53 | 12.87 | 23.44 | 14.88 | 14.15 | 26.05 |
| | STGODE | 12.75 | 13.26 | 21.66 | 14.12 | 14.57 | 24.64 | 15.50 | 16.34 | 27.43 |
| | DSTAGNN | 13.18 | 12.15 | 21.93 | 16.37 | 18.82 | 27.41 | 19.99 | 19.97 | 33.73 |
| | D²STGNN | 12.18 | 12.00 | 21.30 | 13.48 | 13.20 | 24.30 | 14.90 | 14.27 | 27.28 |
| Incident | LSTM | 14.17 | 10.13 | 23.75 | 17.41 | 15.38 | 29.43 | 20.93 | 14.33 | 34.05 |
| | ASTGCN | 14.06 | 10.55 | 23.22 | 16.42 | 15.06 | 27.63 | 18.40 | 12.59 | 30.48 |
| | DCRNN | 13.62 | 9.86 | 23.04 | 15.36 | 14.35 | 26.73 | 16.92 | 11.69 | 29.38 |
| | AGCRN | 14.48 | 10.98 | 25.41 | 15.96 | 14.78 | 28.78 | 17.21 | 11.78 | 31.42 |
| | GWNET | 13.73 | 10.44 | 22.90 | 15.60 | 14.50 | 26.73 | 17.15 | 11.49 | 29.07 |
| | STGODE | 14.50 | 10.71 | 24.49 | 16.20 | 15.19 | 27.69 | 17.55 | 12.29 | 30.17 |
| | DSTAGNN | 14.79 | 10.57 | 24.22 | 18.24 | 17.91 | 30.48 | 21.95 | 15.38 | 35.67 |
| | D²STGNN | 13.73 | 10.05 | 23.30 | 15.51 | 14.22 | 27.29 | 17.03 | 11.46 | 30.23 |

**Results**. As shown in Table 3, **all baselines perform significantly better in predicting on the general test dataset compared to the incident test dataset**, since incidents added irregularity into the traffic systems. This suggests that investigating how to improve the performance of forecasting models on time series prediction following an incident is worthwhile and warrants further research and discussion. More details in Appx. A.8.

Table 4: Performance among the SOTA time series classification methods across the datasets, with the top 1 in grey, 2nd in boldface, and 3rd underlined.

| Methods | speed channel-only | | | occupancy channel-only | | | flow channel-only | | | All channels Mixed | | |
|---|---|---|---|---|---|---|---|---|---|---|---|---|
| | Acc | Precision | Recall | Acc | Precision | Recall | Acc | Precision | Recall | Acc | Precision | Recall |
| DT | 41.6% | 41.5% | 41.5% | **40.4%** | **40.2%** | **40.2%** | **39.4%** | **39.3%** | **39.3%** | **41.6%** | **41.4%** | **41.5%** |
| TS2Vec | 36.6% | 36.2% | 36.2% | 36.6% | 36.5% | 36.4% | 37.3% | 37.0% | 37.0% | 37.3% | 37.0% | 37.0% |
| gMLP | **41.3%** | **41.2%** | **41.1%** | 38.4% | 38.3% | 38.3% | 37.3% | 37.2% | 37.2% | 41.6% | 41.5% | 41.5% |
| Sequencer | 35.8% | 35.8% | 35.6% | 35.6% | 35.3% | 35.2% | 34.1% | 33.9% | 33.9% | 40.3% | 40.2% | 40.2% |
| OmniScaleCNN | 35.7% | 35.1% | 35.1% | 36.9% | 36.3% | 36.3% | 37.0% | 36.8% | 36.8% | 40.9% | 40.8% | 40.8% |
| PatchTST | 38.3% | 38.1% | 38.1% | 39.0% | 38.6% | 38.7% | 39.5% | 39.3% | 39.3% | 39.4% | 39.4% | 39.3% |
| FormerTime | 35.9% | 31.0% | 33.4% | 41.0% | 41.1% | 40.8% | 37.8% | 38.2% | 37.3% | 40.5% | 40.5% | 40.1% |

## 4.3 Incident Classification

Since traffic incidents typically affect the traffic on roads, it is viable to deduce the traffic conditions based on the dynamics of the parameters. In this work, a time series classification task is designed on TraffiDent, which involves inferring incident categories based on the traffic during particular time slots detected by the sensors.

**Experimental setting**. Since traffic sensors are not always available at the site of an incident, we start by identifying the nearest sensor affected by each incident according to the distance (i.e., the ABS PM in Table 6). Then, we extract recorded indexes (traffic speed, lane occupancy, and traffic flow) in these sensors during a time window when the incident occurs. Augmented with normal data, these form the basis for characterizing traffic, which fall into three categories: "accidents", "hazards", and "normal". According to Fig. 3(b), we standardize the duration length as the 95th percentile, i.e., 2 hours (w=24). The task is defined as a multivariate time series classification which uses three-channel time series to infer the situations of the traffic. We selected representative baselines from various families including statistical learning, contrastive learning, sequential models and Transformer-based models. More details of experiments in Appx. A.9.

**Performance Evaluation.** From Table 4: (1) **The best classification can achieve 41% accuracy, indicating classifying traffic conditions based on traffic indexes is feasible (better than random guess).** Among the datasets with different inputted features, *DT* and *PatchTST* always outperform the baselines. *gMLP* also shows strong performance, notably achieving top ranks in several categories and particularly excelling in the mixed channels. (2) **Variability across channels.** Different methods exhibit varying degrees of effectiveness depending on the channel used, e.g., *Sequencer* performing better with speed and worse with flow, and *OmniScaleCNN* opposite. Thus, selecting appropriate features can guarantee the model effectiveness.(3) **Integrating multiple features often leads to better classification.** However, the performance gain is method-dependent. *DT* and *gMLP* show improvement, while *TS2Vec* and *OmniScaleCNN* benefit less.

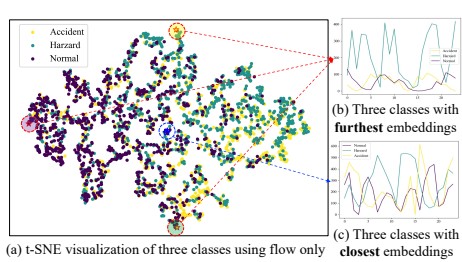

(a) t-SNE visualization of three classes using flow only

(b) Three classes with **furthest** embeddings

(c) Three classes with **closest** embeddings

Figure 4: Visualization of the representation of the time series on the dataset, extracted from the last hidden layer of *OmniScaleCNN*.

Fig. 4(a) visualizes the extracted feature from *OmniScaleCNN* by t-SNE [50] using flow data.The selected furthest embeddings (in Fig. 4(b)) shows clear distinct flows between the three classes, yet the closest embeddings (Fig. 4(c)) not. (1) **There are distinct and separated patterns in the embeddings of traffic incidents located in corner and center areas**, which facilitates classification, causing better performance than random guess. (2) **Not all incidents impact traffic indices largely**. As car fires and accidents with no injuries are short-lived, these hard cases confuse the classifier since the traffic patterns do not change significantly.

## 4.4 Global Traffic Causal Analysis

**Experiment Setting**. In our TraffiDent dataset, we have static variables, e.g., road information, represented as scalar and vector, and dynamic variables, e.g., accidents and traffic flow, represented as functional data. Considering the multimodal nature of the variables, we employ MM-DAG [22] to construct the causal network in different districts. We collect data on 17 variables across four districts with the highest incident rates throughout 2023.

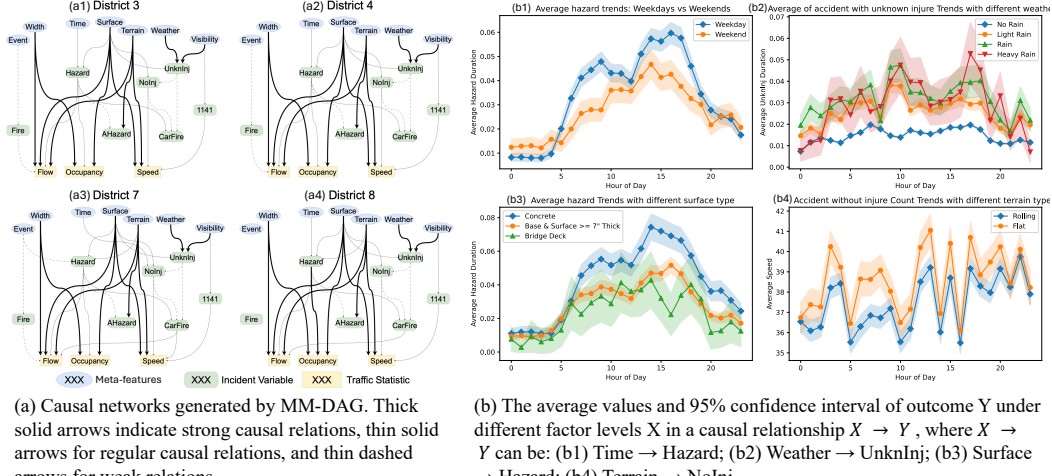

(a) Causal networks generated by MM-DAG. Thick solid arrows indicate strong causal relations, thin solid arrows for regular causal relations, and thin dashed arrows for weak relations.

(b) The average values and 95% confidence interval of outcome Y under different factor levels X in a causal relationship $X \rightarrow Y$, where $X \rightarrow Y$ can be: (b1) Time $\rightarrow$ Hazard; (b2) Weather $\rightarrow$ UnknInj; (b3) Surface $\rightarrow$ Hazard; (b4) Terrain $\rightarrow$ NoInj.

Figure 5: Global causal: (a) The learned DAG among meta-features, incidents, and traffic indexes. (b) The factual explanation for selected edges.

These variables fall into three categories: (1) meta-feature variables, e.g., temporal, environmental, road structural information; (2) incident variables, i.e., the occurrence of incidents; and (3) traffic statistics, which reflect traffic conditions. (Details in Table 7, Appx. A.10). We consider the data collected at each road node for each day as a single sample, using a granularity of one hour.

**Results**. Fig. 5(a) illustrates the four causal networks derived by MM-DAG. (1) **Certain static variables are essential**; such as road surface, terrain, and road width, they exert significant influences on the incidence of traffic events and the overall traffic conditions. (2) **The static variables' impacts are consistent**: Across various districts, due to their inherent properties, these underlying attributes consistently influence road and traffic dynamics across different regions. (3) **Dynamic variables like time, weather, and visibility also affect traffic incidents, though their causal relationships appear to be weaker and vary by district**. Like, causal links from events to fire and to hazards show variability and weaker connections in different districts.

We further explain four significant edges. In Fig. 5(b1), the probability of encountering hazards is higher during the early morning (5 AM to 8 AM) and late afternoon (after 3 PM) on weekdays compared to weekends, likely due to increased traffic flow during peak commuting hours. In Fig. 5(b2), rain increases the probability of accidents compared to dry conditions, but the amount of rainfall does not significantly affect the accident rate. In Fig. 5(b3), bridges and road surfaces with a base thickness of >7 inches have a lower probability of hazards compared to concrete surfaces. This may be due to the enhanced durability and grip provided by thicker road surfaces and bridge constructions. In Fig. 5(b4), flat terrain is associated with higher average speeds compared to rolling terrains. The tendency for higher speeds on flat terrains is likely due to the reduced need for vehicles to decelerate for climbs or curves, allowing for more consistent and faster travel.

The use of causal graph learning for causal analysis is not necessarily aimed at uncovering the absolute "true" causal structure or relationships. For the global causal analysis, we provide case studies to verify whether the discovered causal relationships are consistent with real-world traffic patterns. These factual examples serve to qualitatively validate the reliability of the causal analysis results. For the local causal analysis, the primary value lies in providing useful insights into the underlying data and relationships, which can support better interpretation, hypothesis generation, and downstream decision-making.

## 4.5 Local Causal Analysis for Road Relations

To demonstrate the value of our TraffiDent dataset in revealing the causal relations among the roads, we conduct local causal analysis on a real case from TraffiDent. We employ the PCMCI$^+$ [42] algorithm for causal structure learning. Since the underlying ground truth of causal dependencies is unknown, the hyperparameters, e.g., significance level and maximum time lag, are set for better

interpretability.

**Experiment Setting**. In Fig. 6, we select the road network near an interchange in Novato, California. On the evening of February 11, 2023, a traffic incident "1141" occurred at the eastbound exit of the interchange. This incident indicates that a traffic collision occurred and that there were potential injuries requiring a medical response. We then select four traffic nodes, represented by their traffic flow indexes, $\{X^1, X^2, X^3, X^4\}$, that might be affected by the traffic incident. We see from Fig. 6 (a2) that the decrease in traffic flow of $X^3$, which is the node closest to the incident, accelerated after the incident occurred.

**Results**. The causal graphs learned by PCMCI$^+$ are in Fig. 6 (b), where the colors depict the strength of causal dependencies and the label of a link represents the time lag of causal dependencies. The pre-incident causal structure matches the common understanding about traffic propagation, indicating that traffic flow propagates from $X^2$ through $X^3$ to $X^4$. Compared to the pre-incident graph, the post-incident graph has two additional lagged causal links $X^3 \xrightarrow{\text{lag 1}} X^2$ and $X^4 \xrightarrow{\text{lag 1}} X^1$. In such a complicated dynamic traffic system, explaining the change of causal dependencies is challenging and we endeavour to provide some conjectures for reference. Due to the traffic collision between $X^2$ and $X^3$, congestion likely occurred near $X^3$,

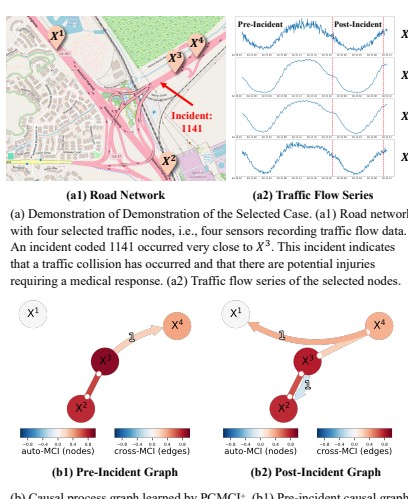

(a) Demonstration of Demonstration of the Selected Case. (a1) Road network with four selected traffic nodes, i.e., four sensors recording traffic flow data. An incident coded 1141 occurred very close to $X^3$. This incident indicates that a traffic collision has occurred and that there are potential injuries requiring a medical response. (a2) Traffic flow series of the selected nodes.

(b) Causal process graph learned by PCMCI$^+$. (b1) Pre-incident causal graph. (b2) Post-incident causal graph. Node and link colors depict the strength of auto-dependencies or cross-dependencies, respectively.

Figure 6: Case of Local Causal Analysis

reducing the traffic flow at each time slot. However, the traffic demand from $X^2$ to $X^3$ did not decrease in the short term, causing the congestion to gradually spread to $X^2$. The congestion at the eastbound exit of the interchange led to a decrease in traffic demand from $X^1$ to $X^4$. Fewer vehicles chose to slow down to enter the ramp, causing increased speed and higher traffic flow at $X^1$. (Details in Appx. A.11.)

## 5    Conclusion

We propose a pioneering traffic and incident dataset TraffiDent. It integrates traffic flow data with incident records and road comprehensive meta-features, filling a significant gap in traffic analysis and Incident analysis. TraffiDent lays a solid groundwork for research focused on understanding traffic dynamics, causality, and interrelationships. Through four groups of experiments, we demonstrate that our dataset offers expanded possibilities for research in traffic forecasting, incident classification, and detection, as well as causal analysis.

## Acknowledgements

This work was supported by King Abdullah University of Science and Technology (KAUST) funding. For computer time, this research used Ibex managed by the Supercomputing Core Laboratory at KAUST in Thuwal, Saudi Arabia.

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

# A Appendix

In this Appendix, we will introduce the data sheet of TraffiDent, the statement of responsibility, more details of TraffiDent data, related work of the 4 traffic tasks, and the experiment settings of the Post-Incident Traffic Forecasting, Incident Classification, Global Causal Analysis, and Local Causal Analysis.

## A.1 Data Sheet of TraffiDent

In this section, we follow the datasheet format [13] to answer the critical questions to a standard dataset.

### A.1.1 Motivation

- **For what purpose was the dataset created?** The TraffiDent is the most recent in terms of the collection period and contains the largest number of sensors, covering three distinct types of traffic volume. This ensures the timeliness of traffic research, providing a robust foundation for studies aiming to capture and explain traffic dynamics, causation, and interrelations. TraffiDent serves as a rigid testing bed and empirical support to justify model effectiveness and interoperability in deep learning and the traffic community.

- **Who created the dataset?** The Machine Intelligence and kNowledge Engineering (MINE) lab.

- **Who funded the creation of the dataset?** The creation of the dataset and research reported in this paper was supported by funding from King Abdullah University of Science and Technology (KAUST).

### A.1.2 Composition

- **What do the instances that comprise the dataset represent.** See the Section 3 Data Introduction.

- **How many instances are there in total?** For traffic time series data, the total number of instances is 105120 (Time Slots Number) $\times$ 16,972 (Sensor Number) $\times$ 3 (Feature Number). For incidents, the instances is 1,441,904.

- **Does the dataset contain all possible instances or is it a sample (not necessarily random) of instances from a larger set?** All possible instances excluding the sensors with a large number of missing values.

- **What data does each instance consist of?** See the Section 3 Data Introduction.

- **Is there a label or target associated with each instance?** See the Section 3 Data Introduction.

- **Is any information missing from individual instances?** Raw data missing.

- **Are relationships between individual instances made explicit** Yes, they are connected by time, location, and sensor ID.

- **Are there recommended data splits?** For Traffic forecasting, we recommend the ratio of 6:2:2 for training, valid, and test dataset. It is a common setting [44, 15].

- **Are there any errors, sources of noise, or redundancies in the dataset?** Yes. The traffic time series are collected from sensors, and it may not count all of the passing vehicles.

- **Is the dataset self-contained, or does it link to or otherwise rely on external resources?** No.

- **Does the dataset contain data that might be considered confidential?** No.

- **Does the dataset contain data that, if viewed directly, might be offensive, insulting, threatening, or might otherwise cause anxiety?** No.

### A.1.3 Collection Process

- **How was the data associated with each instance acquired?** The data is directly observable.

- **What mechanisms or procedures were used to collect the data (e.g., hardware apparatuses or sensors, manual human curation, software programs, software APIs)?** We use the PeMS data table corresponding URLs to collect the data.

- **If the dataset is a sample from a larger set, what was the sampling strategy (e.g., deterministic, probabilistic with specific sampling probabilities)?** Not fit.

- **Who was involved in the data collection process (e.g., students, crowdworkers, contractors) and how were they compensated (e.g.,how much were crowdworkers paid)?** No person is involved in the collection process.

- **Over what timeframe was the data collected?** The data was collected from April 20, 2024, to May 10, 2024. The dataset covers the entire 3 years from 2022 to 2024.

- **Were any ethical review processes conducted (e.g., by an institutional review board)?** No.

### A.1.4 Preprocessing/cleaning/labeling

- **Was any preprocessing/cleaning/labeling of the data done (e.g., discretization or bucketing, tokenization, part-of-speech tagging, SIFT feature extraction, removal of instances, processing of missing values)?**Yes. We construct the adjacency matrix for sensors in the road network.

- **Was the "raw" data saved in addition to the preprocessed/cleaned/labeled data (e.g., to support unanticipated future uses)?** Yes.

- **Is the software that was used to preprocess/clean/label the data available?** The code is released in our GitHub repository https://github.com/xaitraffic/xtraffic

### A.1.5 Uses

- **Has the dataset been used for any tasks already?** No.

- **Is there a repository that links to any or all papers or systems that use the dataset?** No.

- **What (other) tasks could the dataset be used for?** Interpretable traffic forecasting, incident classification, incident duration prediction, and traffic causal analysis.

- **Is there anything about the composition of the dataset or the way it was collected and preprocessed/cleaned/labeled that might impact future uses?** The adjacency matrix is generated based on a threshold. It could be revised based on the task requirement.

- **Are there tasks for which the dataset should not be used?** No.

### A.1.6 Distribution

- **Will the dataset be distributed to third parties outside of the entity (e.g., company, institution, organization) on behalf of which the dataset was created?** Yes.

- **How will the dataset will be distributed (e.g., tarball on website, API, GitHub)?** Kaggle Dataset.

- **When will the dataset be distributed?** June 11, 2024.

- **Will the dataset be distributed under a copyright or other intellectual property (IP) license, and/or under applicable terms of use (ToU)?** Yes. Please see the section 4.

- **Have any third parties imposed IP-based or other restrictions on the data associated with the instances?** Yes. Please see the section 4.

- **Do any export controls or other regulatory restrictions apply to the dataset or to individual instances?** Yes. The use of data is required to satisfy the Caltrans Terms of Use of PeMS.

### A.1.7 Maintenance

- **Who will be supporting/hosting/maintaining the dataset?** The first author of the dataset paper.

- **How can the owner/curator/manager of the dataset be contacted?** After accepting, we will release the email of the owner.

- **Is there an erratum?** No.

- **Will the dataset be updated?** Anual. If someone reports the error to us via GitHub, Kaggle or Email, we will check the data and fix the errors.

- **If the dataset relates to people, are there applicable limits on the retention of the data associated with the instances (e.g., were the individuals in question told that their data would be retained for a fixed period of time and then deleted)?** There is no person information included in TraffiDent.

- **Will older versions of the dataset continue to be supported/hosted/maintained?** Yes. The largest difference between the old version and the new version is the time, and it's not hard to maintain the old versions. we will fix the errors reported.

- **If others want to extend/augment/build on/contribute to the dataset, is there a mechanism for them to do so?** No. We don't have enough resources to verify the external contributions.

## A.2 Statement of Responsibility

According to the Ownership section in Caltrans Terms of Use of PeMS[3], we can collect and construct a dataset from the source and distribute it. We collected all of the data before 17/05/2024. More details and the introduction of the dataset can be found in the supplementary material. Our TraffiDent is released under a CC BY-NC 4.0 International License[4]. The code for the experiments is released under an MIT License[5]. We claim that we are responsible for the data release and collection.

## A.3 Details of TraffiDent

**Licence**. According to the Ownership section in Caltrans Terms of Use of PEMS, we can collect and construct a dataset from the source and distribute it. We collected all of the data before 17/05/2024. More details and the introduction of the dataset are clarified in the Appendix A.3. Our TraffiDent is released under a CC BY-NC 4.0 International License. The code for the experiments is released under a MIT License.

**Meta data**. The meta data for the TraffiDent dataset can be accessed at the `https://github.com/XAITraffic/XTraffic/blob/main/xtraffic-metadata.json`.

---

[3]https://pems.dot.ca.gov/?view=tou
[4]https://creativecommons.org/licenses/by-nc/4.0
[5]https://opensource.org/licenses/MIT

**Incidents**. The details of incident data features are shown in Table 5.

Table 5: Meta Feature Introduction

| Feature | Type | Description |
|---|---|---|
| Incident ID | Integer | Unique identifier for each recorded traffic incident. |
| Duration | Integer | Length of the incident measured in minutes from start to resolution. |
| Abs PM | Float | Point of the incident in absolute postmile notation along the road. |
| Fwy | String | The freeway ID where the incident occurred. |
| AREA | String | The city or town where the incident took place. |
| DESCRIPTION | String | A brief narrative describing the specifics of the incident. |
| LOCATION | String | The exact address on the freeway where the incident happened. |
| Type | String | Category of the incident, such as Noinjure, CarFire, and Hazzard. |
| dt | DateTime | Timestamp indicating when the incident was first reported. |
| Latitude | Float | The latitude where the incident was first reported. |
| Longitude | Float | The longitude where the incident was reported. |
| Direction | String | The direction of the lane where the incident happened. |

**Lane meta features**. The details of the lane meta features are in Table 6,

Table 6: Meta Feature Introduction

| Feature | Type | Description |
|---|---|---|
| Sensor ID | String | Unique identifier for each traffic sensor. |
| Inner Shoulder Width | Float | Width in meters of the inner shoulder on the lane. |
| Outer Shoulder Width | Float | Width in meters of the outer shoulder on the lane. |
| Functional Class | String | Classification of roads based on the function they provide. |
| Inner Median Type | String | Type of median on the inner side of the road. |
| Inner Median Width | Float | Width in meters of the median on the inner side of the road. |
| Road Width | Float | Total width in meters from one side to the other. |
| Lane Width | Float | Width in meters of each traffic lane on the road. |
| Design Speed Limit | Integer | Maximum speed limit designed for the road in kilometers per hour. |
| Terrain | String | Physical features and shape of a landscape, e.g., flat, mountainous. |
| Population | String | Type of terrain surrounding the road, e.g., urban, rural. |
| Barrier | String | Description of any barriers along the road, e.g., guardrail, none. |
| Surface | String | Road surface type, e.g., asphalt, concrete. |
| Roadway Use | String | Primary use of the road, e.g., commercial, residential. |
| Length | Integer | The total length of the lane on the road. |
| Latitude | Float | Geographical latitude of the road's location. |
| Longitude | Float | Geographical longitude of the road's location. |
| Abs PM | Float | Point of measurement in absolute postmile notation along the road. |
| Direction | String | The direction of the lane, e.g., East, North. |
| Fwy | String | The ID of the freeway where the sensor is located in. |
| Fwy Name | String | The name of the freeway where the sensor is located in. |
| District | Integer | The district ID, e.g.. |
| County | String | The county where the sensor is located in, e.g., Orange, Los Angeles. |
| City | String | The city where the sensor is located in, e.g., Marina, Oakland. |
| Sensor Type | String | The sensor cateogry, e.g., radars, magnetometers. |
| Type | String | The level of the road, e.g., mainline, On Ramp. |
| HOV | String | Whether it is HOV lane or not |

## A.4 Preliminary of Four Intended Tasks

**Traffic Forecasting** is to predict traffic indexes at nodes within a road network based on historical data collected by sensors at each node. Consider a traffic road network represented as a graph $G = (V, E)$, where $V$ denotes the set of $N$ traffic nodes, with $|V| = N$, and $E$ represents the set of undirected edges. An edge $E_{ij} = 1$ indicates a physical connection between nodes $i$ and $j$ in the road network; otherwise, $E_{ij} = 0$. Traffic volume data is recorded by sensors in evenly spaced time intervals and can be represented as a sequence of matrices $(\mathbf{X}^1, \mathbf{X}^2, ..., \mathbf{X}^T) \in \mathbb{R}^{N \times T}$, where $\mathbf{X}^t$ is the matrix of volume signals $(x_1^t, x_2^t, ..., x_N^t)$ at time slot $t$ for all $N$ nodes.

The goal in traffic flow forecasting is to devise a function $\mathcal{F}_1$ that uses the observed traffic data from $T_1$ time slots to predict the traffic volumes for the subsequent $T_2$ time slots: $(\mathbf{X}^{t-T_1+1}, ..., \mathbf{X}^t) \xrightarrow{\mathcal{F}_1} (\hat{\mathbf{X}}^{t+1}, ..., \hat{\mathbf{X}}^{t+T_2})$, where $\hat{\mathbf{X}}^{t+1}$ represents the prediction at time $t+1$, and a general loss function is defined as: $\min \frac{1}{T_2} \sum_{i=1}^{T_2} \mathcal{L}_1(\hat{\mathbf{X}}^i, \mathbf{X}^i)$.

**Incident Classification** is to identify traffic incidents using traffic indexes. Since traffic sensors are not always available at the site of an incident, for brevity, we associate the traffic in the nearest single sensor to classify an incident, rather than aggregating data from multiple neighboring sensors. For the $i$-th paired sample $(\mathbf{X}_i, y_i)$ in the dataset $\mathcal{D}$, $\mathbf{X}_i^c \in \mathbb{R}^{C \times w}$ is the input and $y_i$ represents its corresponding label, where $C$ denotes the number of multivariate feature channels (e.g., speed and flow) and $w$ indicates the time window at the post-incident timing $t$. There is $\mathbf{X}_i^c = \{\mathbf{x}^t, \mathbf{x}^{t+1}, ..., \mathbf{x}^{t+w-1}\}$ and the $j$-th entity $\mathbf{x}^{t+j} \in \mathbb{R}^C$. The classification task is: $\hat{y}_i = \mathcal{F}_2(\mathbf{X}_i^c; \Theta)$, where $\hat{y}_i$ is the predicted result, $\mathcal{F}_2$ is the classifier, and $\Theta$ is trainable parameters. The overall objective is to minimize the classification loss $\mathcal{L}_2$ (e.g., cross-entropy) on $\mathcal{D}$: $\min_\Theta \frac{1}{|\mathcal{D}|} \sum_{i=1}^{|\mathcal{D}|} \mathcal{L}_2(\mathcal{F}_2(\mathbf{X}_i^c; \Theta), y_i)$.

**Causal Analysis and Directed Acyclic Graph (DAG)** In causal analysis, the primary objective is to elucidate the causal relationship, which is represented in a dynamic acyclic graph (DAG). Within a DAG, each node corresponds to a variable, and each directed edge delineates a causal relationship between two variables. The causal structural model enables the representation of a node's distribution $w_i$ through $w_i = f_i(w_{pa_i}, e_i)$, where $w_{pa_i}$ denotes the set of all parents of node $w_i$, and $e_i$ represents the exogenous noise associated with node $w_i$. We consider two subtasks:
(1) *Global Causal Analysis for The Whole System*. In global causal analysis, we focus on the problem that macro-level phenomena influence each other, such as the impact of weather on accident rates. Therefore, each node within the graph represents distinct variables like weather conditions, traffic accidents, or overall traffic statistics. This approach helps in understanding the broader implications of various environmental and systematic factors on traffic dynamics.
(2) *Local Causal Analysis for Road Relations*. In local causal analysis, we focus on the temporal dependency structure underlying the complex traffic road network. We aim to find a graph $\mathcal{G}$ where the nodes are the variables representing traffic nodes at different lag-times and the links represent lagged or contemporaneous causal dependencies between traffic nodes. This approach helps in understanding how topologic of the road network affects traffic conditions.

## A.5 Data Cleaning

Besides filling zero and linear interpolation, there are also some specific filling models for traffic data. For example, ST-MVL[6] combined several empirical statistical models with user-based and item-based collaborative filtering to collectively fill in missing values in geo-sensory time series data. However, the model suffers from overlooking the global correlations of data. The [7] regards the raw data as a tensor and models the data recovery as a low-rank robust tensor completion via leveraging the inherent low-rank structure to address the issue. On the other hand, to discover the anomaly/dirty/outlier data, the [8] leverages DBSCAN to discover the outlier points in spatio-temporal data. Furthermore, it's necessary to repair the discovered dirty data. [9] proposes a metric to evaluate the dissimilarity between the raw dataset and the repaired one. Then it utilizes space- and time-distortion rules and employs a hybrid simulated-annealing approach to avoid local minima during the repair process. We will add the discussion of potential data cleaning methods in the final paper or on the dataset project website. Exploring better data-filling techniques to mitigate the impact of data gaps is an excellent direction for future research.

### A.6 Construction of Adjacency Matrix

Typically, the adjacency matrix is constructed based on distance [32]. In order to get the real travel distance, we set up an open-source rooting machine engine [35] based on OpenStreetMap, and calculate the shortest travel distance between two sensors based on the coordinate. One more precise adjacency matrix is constructed based on the direction of the lanes and the coordinates of two sensors $A$ and $B$. To ensure directional consistency in the spatial graph, we exclude neighboring sensors located on roads with opposite traffic directions. Since the meta features of each sensor already encode its directional attribute, the inclusion of reverse-direction neighbors would introduce redundancy. For example, if a sensor is positioned on a northbound road, its southbound counterparts are removed from the adjacency set, while neighbors located along the west, east, and north directions are retained.

### A.7 Definition of Hub and Fringe Nodes

We calculate the degree of each node using the adjacency matrix described in the paper. Nodes with the 500 highest degrees are classified as hub nodes, and those with the 500 lowest degrees are classified as fringe nodes. We match all incidents with the closest node/sensor and also remove the incident samples that the distance between the incident and its closest sensor is larger than 0.05 mile. We count the incident number of the hub nodes and fringe nodes, respectively.

### A.8 Experiment Details on Post-Incident Traffic Forecasting

**Baselines**. The baselines we selected to do the forecasting experiments are typical models in traffic forecasting domain.

- **LSTM**[17]: A basic model focusing solely on the temporal relationships within traffic data.
- **ASTGCN**[15]: Enhances the STGCN by incorporating an attention mechanism to better capture node correlations.
- **DCRNN**[27]: An RNN-based model that utilizes diffusion convolution to model traffic flows.
- **AGCRN**[4]: An adaptive model that combines RNN architecture with an attention mechanism to focus on spatial correlations.
- **GWNET**[51]: Utilizes a gated mechanism in a TCN framework to filter out irrelevant information effectively.
- **STGODE**[12]: Uses ordinary differential equations to dynamically model relationships among traffic nodes.
- **DSTAGNN**[21]: Designed to dynamically capture changing correlations among traffic sensors.
- **D$^2$STGNN**[44]: A dual-layer spatial-temporal GNN that addresses hidden correlations in traffic data for forecasting.

**Implementation Details**. We adhered to the identical experimental settings outlined within the work. We divided all the data into training, validation, and test sets in a 6:2:2 ratio. We set the batch size as 24 for DSTAGNN and 64 for all of other models. The learning rate is set as 0.001. Other hyperparameters of models are set as the same as the original settings. Our baselines follow the optimal settings from their sources. For the batch size in the training set, we used different settings to ensure that the model converges as quickly as possible during training. The batch size settings are mentioned in Section 4.2. For **LSTM**, the hidden layer dimension is set as 64, the last linear layer dimension is set as 512. For **ASTGCN**, the dimension of the attention layer is as 64. For **DCRNN**, the number of RNN layers is set as 2, and the dimension for each RNN layer is 64. For **AGCRN**, the hidden dimension is set as 64 for all cells and the embedding dimension is set as 10. For **GWNET**, the dimension of input and output linear layer are set as 32 and 512, respectively. The dimension of hidden layers is set as 256. For **STGODE**, the regular hyperparameter $\alpha$ is set as 0.8. The thresholds $\sigma$ and $\epsilon$ of spatial adjacency matrix (AM) are set to 10 and 0.5 respectively, and the threshold $\epsilon$ of the semantic AM is set to 0.6. For **DSTAGNN**, the attention dimension is set as 32, and the number of attention heads is set as 3. For **D$^2$STGNN**, the hidden dimension is set as 32.

**Traffic and incident Alignment**. To explain which types of traffic are affected by incidents, we perform the following traffic–incident alignment. The traffic and incident data in TraffiDent can be spatially aligned using geographical coordinates (longitude and latitude) and absolute postmile (abs PM). The features of sensors measuring traffic and incident samples all include these locations. For temporal alignment, we provide: Timestamps for all incident records, and 5-minute interval traffic time series, starting from 2022-01-01 00:00. Users can derive the time slot of any traffic measurement using the known start time and interval length.

In the traffic forecasting experiment (Section 4.2), we provide the following alignment steps.

(1) For each reported incident in one specific area (In the experiment introduced in this section, it is D5 (Monterey)), we identify the corresponding freeway. (2) We match the incident with the closest sensor (e.g., Sensor A) on the same freeway based on the Absolute Postmile (Abs PM). (3) The incident timestamp is converted into a 5-minute time slot (e.g., 19:13 becomes 19:10–19:15), and the next slot (e.g., 19:15–19:20) is defined as the starting point of post-incident influence. (4) Given the multistep nature of our forecasting task, we select the traffic from 19:15 to 19:45 at Sensor A as a single post-incident test case (Using traffic from 18:15 to 19:15 as input feature).

We select all post-incident samples based on the above (2)-(4) steps.

## A.9 Experiment Details on Incident Classification

**Baselines.** We adopt the following representative time series classification baselines.

- **Decision Tree (DT)**: We tailor the canonical decision tree algorithm for the task, recursively partitioning data based on feature values to create a tree-like model that makes classifications at its leaf nodes.
- **TS2Vec** [57]: It is a universal framework for learning robust and flexible time series representations using hierarchical contrastive learning over augmented context views, making the classification by a linear classifier.
- **gMLP** [31]: It is a simple network architecture based solely on MLPs with gating, which performs as well as Transformers in key language and vision applications.
- **Sequencer** [49]: It models long-range dependencies using LSTMs without self-attention layers, which enhances performance by reducing the sequence length and creating spatially meaningful receptive fields.
- **OmniScaleCNN** [48]: It is a 1D-CNN architecture that utilizes a set of prime number-based kernel sizes to efficiently capture optimal receptive field sizes without scale tuning across diverse time series classification tasks.
- **PatchTST** [37]: It incorporates patching of time series into subseries-level patches and channel-independence to improve long-term forecasting accuracy based on the Transformer backbone.
- **FormerTime** [6]: It employs a hierarchical Transformer-based architecture to learn multi-scale feature maps and introduces an efficient temporal reduction attention mechanism and a context-aware positional encoding generator for multivariate time series classification.

**Implementation Details**. We randomly sampled 9,000 examples to experiment, 3,000 samples per category. The data is divided into training and testing sets in a 7:3 split.

We set the hyperparameters based on the recommended values in the original method and adjust them around those values, taking the parameter values corresponding to the best results as the final result. Specifically, the hyperparameters for each method are as follows:

For the **Decision Tree**, the minimum number of samples required for a leaf node is set to 1, and for splitting an internal node, it is set to 2. In **TS2Vec**, the pretraining stage has an output dimension of 320 and a hidden dimension of 64. The model is trained for 100 epochs with a batch size of 16, and the linear layer is chosen as the downstream classification module with $1e^{-3}$ learning rate. For **FormerTime**, the model is configured with 3 stages, each having 2 layers with a hidden size of 64. The number of slices per stage is 4, 2, 2, with a stride of 4, 2, 2. The model is trained for 100 epochs with $1e^{-3}$ learning rate. In **PatchTST**, the patch length is set to 16, the stride to 8, the number of encoder layers to 2, the number of heads to 8, and the model dimension to 512. **gMLP** is configured

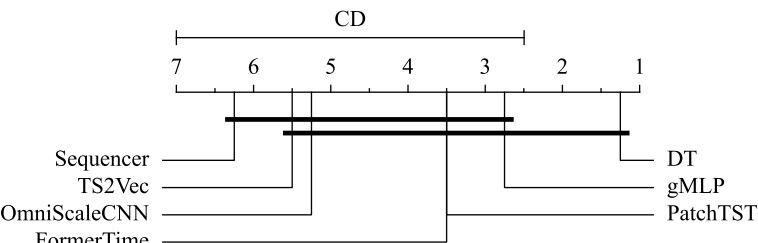

Figure 7: Critical difference diagram over the mean ranks of the compared methods

with a patch size of 1, a model dimension of 256, a fully forward network dimension of 512, and a depth of 6. **PatchTST**, **TSSequencer**, **OmniScaleCNN**, and **gMLP** are trained for 100 epochs with a batch size of 128, and the learning rate of them is set as $3e^{-4}$.

**More Results**. Fig. 7 reports the critical difference diagram as presented in [9], which compares the mean ranks of the baseline methods on the four datasets (three channel-only and all mixed) in the classification task. The thick horizontal lines in the diagram denote groups of methods whose performance differences are not statistically significant within the critical difference (CD) threshold. It can be seen that DT, gMLP, and PatchTST are among the top-performing methods with the lowest mean ranks, indicating their superior performance. Although DT, gMLP, and PatchTST are highlighted as top performers, the differences among the top five methods are not statistically significant since they are in a group, suggesting comparable effectiveness in this task.

### A.10   Experiment Details on Global Causal Analysis

**The introduction of MM-DAG**. MM-DAG is a score-based causal discovery algorithm. It learns multiple DAGs with multimodal data where their consensus and consistency are maximized. For multimodal data, it proposes a multi-modal regression for linear causal relationship description of different variables by functional principal component analysis. For multitask learning, it uses causal difference to ensure the consistency. The overall optimization problem can be represented as:

$$\hat{\mathbf{C}}_{(1)}, ..., \hat{\mathbf{C}}_{(L)} = \underset{\mathbf{C}_{(1)},...,\mathbf{C}_{(L)}}{\arg\min} \sum_{l=1}^{L} \frac{1}{2N_l} \|\mathbf{A}_{(l)} - \mathbf{A}_{(l)}\mathbf{C}_{(l)}\|_F^2$$

$$+ \rho \sum_{l_1,l_2} s_{l_1,l_2} DCD(\mathbf{W}_{(l_1)}, \mathbf{W}_{(l_2)}) + \lambda \sum_{l=1}^{L} \|\mathbf{C}_{(l)}\|_1$$

$$\text{s.t. } h(\mathbf{W}_{(l)}) = \text{tr}(e^{\mathbf{W}_{(l)}}) - P_l = 0, \forall l$$

where $\mathbf{A}$ is variables after FPCA, $\mathbf{C}$ are causal matrix and $\mathbf{W}_{(l)ij} = \|\mathbf{C}_{(l)ij}\|_F^2$. $s_{l_1,l_2}$ is the given constant reflecting the similarity between tasks $l_1$ and $l_2$, $\rho$ controls the penalty of the difference in causal orders, where larger $\rho$ means less tolerance of difference. $\lambda$ controls the $L_1$-norm penalty of causal matrix which guarantees that edges are sparse. In our setting, we set $\lambda = 0.001$, $\rho = 1$ and $s_{l_1,l_2} = 1, \forall l_1, l_2$.

**Explaination of the nodes**: The details of the nodes in the global causal graph in listed in Table 7.

**Constraints of the experiment**:In learning DAG, two constraints are placed: (1) edges are not allowed to point toward meta-feature variables since meta-feature variables generally describe environmental and infrastructural contexts that inherently influence other variables rather than being influenced by them. (2) traffic statistics variables are restricted from directing edges towards other nodes since these statistics fundamentally represent outcomes or states of the traffic system, typically influenced by both high-level environmental conditions and specific incidents, rather than serving as direct causes themselves.

Table 7: The description and types of the variables used in traffic causal analysis

| Category | Name | Type | Description |
|---|---|---|---|
| high-level variables | Time | Scalar | Day type indicator: $= 1$ if the day is a weekend; $= 0$ otherwise. |
| | Event | Scalar | Public holiday indicator: $= 1$ if the day is a public holiday; $= 0$ otherwise. |
| | Visibility | Functional | An integer ranged from 0 to 16 to indicate the visibility of the road in the district. |
| | Surface | Vector | Attributes describing road material, e.g., concrete, bridge deck. |
| | Terrain | Vector | Characteristics of the terrain surrounding the road, e.g., flat, rolling. |
| | Width | Scalar | The width of the road. |
| | Weather | Functional | An integer ranged from 0 (No rain) to 3 (Heavy rain). |
| incident variables | Hazard | Functional | Details of any hazards present, e.g., obstacles, spillage. |
| | NoInj | Functional | Records of accidents with no injuries. |
| | UnknInj | Functional | Records of accidents with unknown injury statuses. |
| | 1141 | Functional | Records of accidents needing an emergency response (coded 1141). |
| | Fire | Functional | Incidents involving vehicle fires or roadside fires. |
| | AHazard | Functional | Presence of animals on the road that could cause hazards. |
| | CarFire | Functional | Specific incidents involving car fires. |
| traffic statistics | Flow | Functional | Measures of traffic flow, typically in vehicles per hour. |
| | Occupancy | Functional | Percentage of the road occupied by vehicles at a given time. |
| | Speed | Functional | Average speed of traffic flow. |

## A.11  Experiment Details on Local Causal Analysis

In local causal analysis, We employ the PCMCI$^+$ algorithm to discover the causal relations in traffic data, which utilizes momentary conditional independence (MCI) test to determine the existence of causal links. Typically, the lagged and contemporaneous causal relations are displayed in a dynamic Bayesian network (DBN) as shown in Fig. 8 (a). In this work, to simplify the visualization, we choose to use the process graph as shown in Fig. 8 (b) to aggregate the information in the DBN. In both DBN and process graph, the link color denotes the magnitude of the causal effect measured by the MCI test statistic (e.g., the partial correlation coefficient). The label of a link lists all significant lags of cross-dependencies in process graph. Since we are more interested in the causal links between different traffic nodes, the links denoting auto-dependencies in DBN are summarized into node colors in process graph and the auto-dependency lags are omitted.

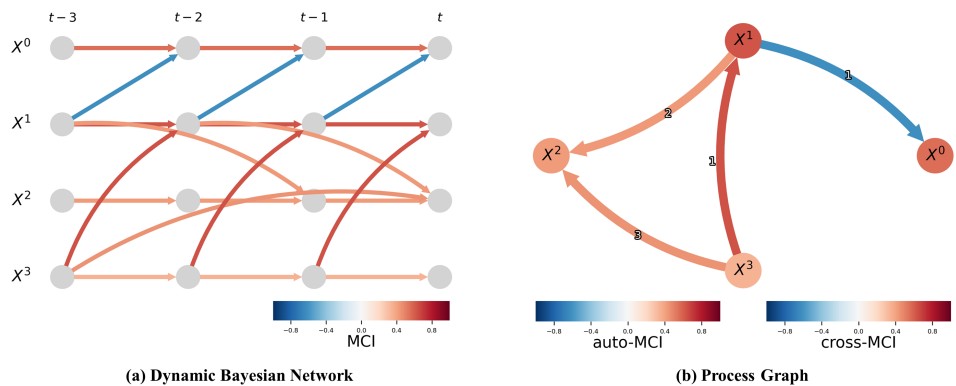

(a) Dynamic Bayesian Network  (b) Process Graph

Figure 8: Examples of causal graphs for time series variables. (a) Dynamic Bayesian network (DBN). (b) Process graph which summarizes the information in DBN. Process graph aggregates the information in the DBN to simplify the causal structure visualization. While the process graph is nicer to look at, the time series graph better represents the spatio-temporal dependency structure from which causal pathways can be read off. In both graphs, link colors depict the magnitude of the cross-node causal effects as measured by the MCI test statistics. In process graph, node colors depict auto-dependency strength.

The choice of causal structure learning method influences the results of local causal analysis. Ideally, we would like to perform analysis on real cases or datasets with known underlying ground truth of causal dependencies. However, such cases or datasets are rare especially in complex dynamic scenarios such as traffic. To enhance the credibility of the learned causal structure, we use different causal discovery methods and verify the consistency of the results obtained by the different methods.

Fig. 9 shows the pre-incident causal graphs of case I learned by score-based method DyNotears [40] and constrained-based method PCMCI$^+$ [42]. The graph structures learned by both methods are similar, but the time lag of the link $X^3 \to X^4$ is different, which is greatly influenced by the sampling frequency of traffic data. Due to the limited number of samples affected by the incidents, we use PCMCI$^+$ to discover post-incident causal structure for its robustness with small sample size and high dimensionality.

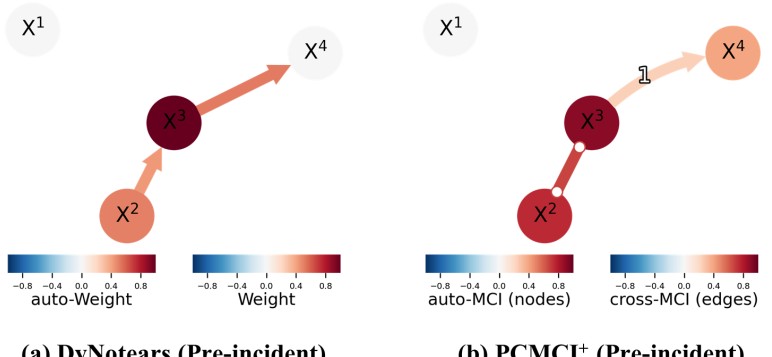

**(a) DyNotears (Pre-incident)**          **(b) PCMCI$^+$ (Pre-incident)**

Figure 9: Pre-Incident Causal Graphs for Local Causal Analysis Case Learned by Different Methods. (a) Process Graph Learned by DyNotears. (b) Process Graph Learned by PCMCI$^+$.

The primary Hyperparameters of $PCMCI^+$ are the maximum time delay $\tau_{\max}$ and the significance threshold for the MCI test $\alpha_{PC}$. The maximum time delay should be determined based on the specific application, reflecting the maximum expected causal time lag in the scenario under investigation. To identify this maximum time lag, we plot the results of the bivariate lagged conditional independence test. The significance threshold $\alpha_{PC}$ is adjusted on a case-by-case basis to ensure the derived causal structure is reasonable for the analysis. Further details about $PCMCI^+$ implementation and parameter tuning can be found in the public causal discovery tutorials [6]

### A.12    Incident Information–Enhanced Traffic Forecasting Test

To examine the influence of incident information on traffic forecasting performance, we extend the STID[43] model by incorporating incident-related features. We follow the standard experimental configuration with 12 historical and 12 forecasting time slots. The number of nodes is 8,614 (only main road sensor). Both the original STID model and the incident-enhanced variants are implemented based on the official STID repository.

We introduce an additional feature dimension representing the presence of incidents. Traffic samples affected by incidents are labeled with an index of 1, and all others are labeled as 0. Following the same criteria as in the main experiments, if a sensor is the closest one to an incident, its traffic data are considered incident-related within the subsequent 10 minutes.

We evaluate the baseline STID model and two variants that integrate incident information:

1. **STID + Incidents (32ED):** uses an embedding hidden size of 32.
2. **STID + Incidents (16ED):** reduces the feature embedding size to 16 to maintain comparable final embedding dimensions.

The experimental results are summarized in Table 8.

Overall, incorporating incident information leads to consistent improvements over the original STID model. The **STID + Incidents (32ED)** variant achieves better performance across all metrics, while the **16ED** configuration further reduces MAE and RMSE by maintaining balanced feature embedding dimensions. Notably, in long-horizon forecasting (e.g., 60 minutes), incident-aware models exhibit a clear performance advantage. However, as traffic incidents are relatively sparse, the direct use of incident indicators provides limited overall gains. These findings suggest that developing **dedicated models to capture incident-driven traffic dynamics** is a promising direction for future research.

---

[6]https://github.com/jakobrunge/tigramite/blob/master/tutorials/causal_discovery/

Table 8: Performance of STID and incident-enhanced variants.

| Horizon | Model | MAE | RMSE | MAPE (%) |
|---|---|---|---|---|
| Average | STID (32ED) | 13.11 | 23.98 | 23.32 |
| | STID + Incidents (32ED) | 13.01 | 23.71 | 22.81 |
| | STID + Incidents (16ED) | 12.69 | 23.32 | 23.55 |
| 1 (5 min) | STID (32ED) | 10.50 | 18.69 | 20.73 |
| | STID + Incidents (32ED) | 10.32 | 18.52 | 18.44 |
| | STID + Incidents (16ED) | 10.29 | 18.34 | 21.37 |
| 6 (30 min) | STID (32ED) | 13.10 | 24.07 | 22.03 |
| | STID + Incidents (32ED) | 13.01 | 23.82 | 23.25 |
| | STID + Incidents (16ED) | 12.63 | 23.41 | 22.72 |
| 12 (60 min) | STID (32ED) | 15.37 | 27.85 | 27.70 |
| | STID + Incidents (32ED) | 15.09 | 27.35 | 25.31 |
| | STID + Incidents (16ED) | 14.63 | 26.76 | 25.73 |

