# OpenReview forum: "TraffiDent: A  Dataset for Understanding the Interplay Between Traffic Dynamics and Incidents"
_NeurIPS.cc/2025/Datasets_and_Benchmarks_Track — NeurIPS 2025 Datasets and Benchmarks Track poster_

### Official Review · Reviewer_PJcj · 2025-06-23

**Rating:** 4
**Confidence:** 5

**Summary:**

This paper proposes a large-scale dataset that aligns traffic time series with incident information, points out the shortcomings of existing traffic spatiotemporal datasets, and explains its own advantages. Subsequently, the paper conducts a detailed statistical analysis of the proposed dataset, including the general characteristics of traffic time series and the special characteristics contained in incident data. Finally, the paper looks forward to the usable scenarios of the proposed dataset and conducts corresponding experimental analysis. The appendix contains the specific details of the dataset and detailed experimental settings. In general, this paper provides more abundant external information for traffic time series analysis - incidents (whose impact on traffic dynamics is direct and strong), which helps users to explore deep causal relationships, build more accurate and realistic prediction models, and further help the construction of intelligent transportation systems and smart cities. However, this paper also has some shortcomings that cannot be ignored, see Limitations Weaknesses for details.

**Additional Feedback:**

I have great expectations for introducing strongly correlated external information to enhance the ability to predict traffic or spatiotemporal sequences, but there is an inherent habit in the field of predicting the future based entirely on historical sequences, and there is a lack of ideal relevant data sets. This paper undoubtedly made a breakthrough attempt, but there are still some aspects that need to be improved and perfected.

**Dataset Code Accessibility:**

Partly

**Dataset Code Comments:**

The dataset is complete, but does not provide the code or detailed tutorials for the data preparation, adjacency graph construction, spatiotemporal alignment process, and experimental testing of the post-accident traffic prediction part.

**Ethical Considerations:**

No, there are no or only very minor ethics concerns

**Final Justification:**

Overall, this paper's proposal to combine traffic state and event data is well-motivated and provides an appropriate data analysis process. The authors diligently addressed constructive criticism, demonstrating a positive attitude and rigorous scientific approach. While the incomplete codebase makes it difficult to conclude that the dataset is highly usable for downstream task validation, the authors have pledged to complete the codebase, and I trust them. Therefore, I am inclined to accept the paper and look forward to its revised form promoting relevant research in the transportation field.

**Limitations Weaknesses:**

1. Uncertain and unclear descriptions of incidents data: 1)Figure 1(b). "1141, Fire NoInjure, UnknownInjure, Hazard, AnimalHazard, CarFire", 2)Figure 3(a). "NoInj, UnknInj, 1141, Fires&Car Fires, Animal, Traffic Hazard", 3)Table 5. "Category of the incident, such as accident, hazard, or road closure", 4)incidents_y2023 "1141, AHazard, CarFire, Fire, Hazard, NoInj, Other, UnknInj", 6)line 185. "TraffiDent covers 7 specific incident types, such as hazzards and road closures", 6)line 209. "The reserved incident data includes 1,441,904 samples with 9 features.". They are not consistent, which is confusing. It is recommended that the author unify the naming to strengthen standardization.

2. Incident data and sensor metadata are difficult to use directly, and it is currently difficult to see whether traffic dynamics and incident data are "spatiotemporally aligned". It is recommended that the author provide processing code or detailed guidance tutorials.

3. The paper is a bit backwards, it uses a lot of space for downstream task experiments, but does not do enough analysis and explanation of the data itself (I can't find a further specific explanation of the various incidents in the paper, and I intuitively wonder if Fire includes CarFire?). If you don't have a clear understanding of the data itself, let alone use it! It is recommended that the author provide a detailed explanation of the meaning of the data, and then use it as a basis for experimental analysis. At present, the paper seems to be a crude use of the data and then a set of analysis procedures.

4. The experiments on traffic time series prediction in the paper are not sufficient and reasonable. 1) Although the spatial and temporal range of the dataset is large, it is reduced during testing, which cannot reflect the advantages; 2) The prediction model used is not comprehensive and novel, and it is difficult to call it line 287. "All of the baselines are state-of-the-art in the spatial-temporal forecasting or traffic forecasting domain.". 3) The process of constructing the adjacency matrix is ​​not clear enough. Appendix A.6 Construction of Adjacency Matrix points out a more precise construction method, but there is no specific implementation guidance or code. 4) line 290. "As shown in Table 3, all baselines perform significantly better in predicting on the general test dataset compared to the incident test dataset" is not accurately described. In fact, the MAPE of 5 Mins shows the opposite result, and there is no explanation in the paper. 5) The paper only screens and experiments data based on incidents, which is not very helpful for incident-driven traffic dynamic analysis, and is far from the ideal introduction of incident information to enhance the prediction effect. Therefore, it is recommended that the authors redesign the experiment and preferably provide a test plan and results for using incident data to assist traffic prediction, which is also the core contribution of this article.

5. line 319. In the incident classification task, the highest accuracy is only 41%, which is close to random guessing. It is recommended that the author re-evaluate data accuracy, experimental settings, and model selection.

6. Some details, not very critical but affect the reading experience. 1) Line 62. "offer a rich collection of physical and policy-level road meta-features" & line 64. "incorporating rich, fine-grained attributes" seem to be just two different expressions of the same concept, and it is inappropriate to divide them into two points of contribution. Similar repetition also appears in line 154-157. & line 164-170. 2) Table 1. "Granules" seems to be "Granularity". 3) The labels "weekdays" and "weekends" in Figure 2 (a) and (b) seem to be reversed. 4) The paper lacks the description of area d in Figure 2 (c) and (d), and there is no analysis and explanation of the special situation of two peak hours in the afternoon due to different areas. 5) Line 306. "parameters" seems to refer to traffic flow, speed and occupancy, which is easily confused with "model parameters". 6) Some elements of Figure 1, Figure 2, Figure 3, and Figure 5 (a) are not vectors, which affects clarity. 7) The order of Figures 4 and 5 is reversed. The authors are advised to review the above issues and make corrections.

**Strengths Contributions:**

The most outstanding contribution of this paper is the introduction of strongly correlated external information, incidents, into traffic time series data. Because the commonly used data sets(PEMS0X, LargeST...) in the field of traffic or spatiotemporal time series forecasting do not contain exogenous variables that are spatiotemporally aligned with endogenous variables, and it is precisely this information that seriously affects the state of the spatiotemporal dynamic system and is an indispensable part of approaching the prediction boundary. The motivation of the paper is clear and the discussion of existing work is thorough and complete. The data and experimental analysis in this paper are relatively complete, and provide relatively rich charts to help readers understand the data characteristics. For example, traffic patterns, incidents proportions, duration distribution, and their relationship with road types and cycle positions. In addition, this paper conducts a preliminary exploration of the relationship between traffic dynamics and incidents, including experiments such as prediction, classification, and causal analysis, which helps readers further understand the complex relationship between the two types of data. Despite the above contributions, this paper does not pay enough attention to the interaction between traffic dynamics and incidents, please refer to the Limitations and Weaknesses section for details.

---

> ### Author Rebuttal · Authors · 2025-07-31
>
> We sincerely appreciate your detailed and insightful suggestions on our work.
> According to the rebuttal rule, we can not update the new figures and external links. We keep updating the new results in the dataset introduction website included in the Abstract.
> ## W1 Uncertainty description
> We clarify that TraffiDent includes **7  incident categories**: 1141 (representing incidents with injuries), Fire, NoInjure, UnknownInjure, Hazard, AnimalHazard, and CarFire. In Figure 1(b), we have revised the labels to match this naming convention. In Figure 3(a), due to the low frequency of Fire and CarFire, we grouped them under a single combined category for visualization purposes and have updated the caption and explanation accordingly. In Table 5, we corrected the incident descriptions in line with the Noinjure, CarFire, and Hazzard categories. In Section 3.1 and Section 3.2, we have ensured consistent usage of incident category names and clarified that the total number of incident records is 1,441,904, as mentioned in line 209.
>
> ## W2 Spatialtemporal alignment
> To clarify, the traffic and incident data in TraffiDent can be spatially aligned using geographical coordinates (longitude and latitude) and absolute postmile (abs PM). The features of sensors measuring traffic and incident samples all include these locations. For temporal alignment, we provide: Timestamps for all incident records, and 5-minute interval traffic time series, starting from 2022-01-01 00:00. Users can derive the time slot of any traffic measurement using the known start time and interval length.
>
> In the traffic forecasting experiment (Section 4.2), we provide the following alignment steps.
>
> 1. For each reported incident in one specific area (In the experiment introduced in this section, it is D5 (Monterey)), we identify the corresponding freeway.
> 2. We match the incident with the closest sensor (e.g., Sensor A) on the same freeway based on the Absolute Postmile (Abs PM).
> 3. The incident timestamp is converted into a 5-minute time slot (e.g., 19:13 becomes 19:10–19:15), and the next slot (e.g., 19:15–19:20) is defined as the starting point of post-incident influence.
> 4. Given the multistep nature of our forecasting task, we select the traffic from 19:15 to 19:45 at Sensor A as a single post-incident test case. (Using traffic from 18:15 to 19:15 as input feature)
> 5. We select all post-incident samples based on the above 2-4 steps.
>
> ## W3 Lack of description
>
> To clarify the description of incidents, we introduce the meaning of the 7 incident categories, covering a wide range of real-world road events:
> - **1141 (Traffic Accident with Injury)**. A traffic accident in which someone is reported as injured.
> -  **NoInjure (Traffic Accident without Injury)**. A traffic accident in which **no injuries** are reported.
> - **UnknownInjure (Traffic Accident, Injury Status Unknown)**. A traffic accident where the report does **not specify** whether any injuries occurred.
> - **AnimalHazard**. An incident caused by animals.  For example, a dog crossing the road causes a vehicle to crash or suddenly brake.
> - **CarFire**. A fire incident where a **vehicle catches fire**, typically due to mechanical or electrical failure.
> - **Fire**. A **non-vehicle fire**, such as a **wildfire** or **lightning strike** igniting a roadside object (e.g., trees or brush).
> - **Hazard**. A hazard caused by **natural or environmental factors**, such as **tornadoes**, **severe thunderstorms**, or other extreme weather events.
>
> Also, we do many analyses from both traffic and incident aspects in Section 4.1. If you think any analysis is critical, we kindly request that to give us a detailed suggestion.
>
> ## W4 Shortcoming of experiments
> - (1) Limited to our computing resources, we cannot finish the experiments on the entire datasets. We extensively implement the experiments in 2 more areas (Fresno and Merced), and we update the results on the main website.
> - (2) There are over 1000 traffic forecasting models proposed in the previous several years; it is not possible to include all.  If we lack a model you think is critical, we could add it to our experiment.
> - (3) To define the graph topology, the common practice [1] is to build the adjacency matrix **A** using a **thresholded Gaussian kernel** [2], defined as:
> $$
> A_{ij} =
> \begin{cases}
> \exp\left(-\frac{d_{ij}^2}{\sigma^2}\right), & \text{if } \exp\left(-\frac{d_{ij}^2}{\sigma^2}\right) \geq r \\
> 0, & \text{otherwise}
> \end{cases}
> $$
> where:
> - $d_{ij}$ is the road network distance between sensors $i$ and $j$,
> - $\sigma$ is the standard deviation of all distances,
> - $r$ is the sparsity threshold.
>
> To construct the adjacency matrix, we use a widely used method in the field relying on road network distances. We utilize the Open Source Routing Machine, a routing engine built on OSM, to compute the shortest driving distances between sensors based on coordinates.
>
> Calculating pairwise distances for a large-scale graph is extremely time-consuming. We first compute pairwise geodesic distances between sensors, which is significantly faster. We then restrict each node to query road network distances only to other nodes within a 4-kilometer radius. We normalize the adjacency matrix following [3] by applying a threshold (0.01) to eliminate weak node connections.  This threshold is flexible and is adjusted by users according to their specific requirements.
>
> [1] Bing Yu, etc. Spatio-temporal graph convolutional networks:
> A deep learning framework for traffic forecasting. IJCAI, 2018.
>
> [2] David I Shuman, etc. The emerging field of signal processing on graphs: Extending high-dimensional data analysis to
> networks and other irregular domains. IEEE Signal Processing Magazine, 2013.
>
> [3] Yaguang Li, etc. Diffusion convolutional recurrent neural network: Data-driven traffic forecasting. ICLR, 2018.
>
> - (4) Due to sensor-based nature, we associate each incident with its nearest sensor. In some cases, the nearest sensor may still be relatively far from the actual incident location, potentially weakening observed impact. Additionally, incidents may occur toward the end of a given time slot—for example, an incident occurring at 17:14 would fall into the 17:10–17:15 time slot for forecasting. As a result, the incident may influence traffic conditions more in the subsequent slots. This is also reflected in our results: at horizons 𝑡 = 15 t=15 and 𝑡 = 30 t=30, MAPE in post-incident scenarios is higher than in general cases, suggesting that the incident’s effect becomes more prominent shortly after its occurrence.
>
> - (5) Leveraging incident information to enhance the prediction is a valuable research direction based on TrafiDent. However, it's not the core contribution of our work. The primary motivation of doing the traffic forecasting experiments is to highlight the shortcomings of existing models in handling post-incident traffic forecasting, and to emphasize the importance of developing new models using datasets that combine traffic incidents, such as TraffiDent. I need to clarify that this is a dataset paper, but not a research track paper.
>
> ## W5 Incident Classification Low Accuracy.
> The original incident labels consist of **7 categories**, but their distribution is **highly imbalanced**, with a severe long-tail pattern. As shown in Figure 3(a), the *“hazard”* category accounts for **52.2%** of all samples, while the combination of the two least frequent categories is **less than 5%**.
>
> Directly performing time series classification under this setting causes models to degrade into **majority-class predictors**, resulting in superficially high accuracy with **no practical utility**. We merged semantically similar labels into two broader categories: **hazard** and **accident**, with a new class ratio of approximately **1.1:1 (52.2%: 47.8%)**.  Under a three-class balanced setting, the **baseline accuracy for random guessing is 33.3%**. Although our achieved accuracy of **41%** may appear modest,  **the precision for each class exceeds 33%**.   For example, when using a decision tree (DT) model for flow channel classification: Hazard: **41.96%**; Incident: **37.96%**; Normal: **37.88%**.
>
> This indicates that the model is not merely favoring the majority class but is indeed learning **distinct and meaningful signal patterns**.  The observed 41% accuracy highlights the inherent difficulty of modeling **incident evolution** using only current time-series features and conventional sequence classification models.  We view this not as a limitation, but as the **motivation of TraffiDent**: we release a realistic and imbalanced dataset to encourage the community to explore more generalizable incident classification/detection methods.
>
> The accuracy reflects the real-world complexity of the problem and the current limitations of existing methods, rather than a flaw in experimental design.
>
> ## W6 Typos and Misunderstandings.
>
> (1) Line 62 Contribution (2) emphasizes that TraffiDent offers a significantly richer set of road meta-features compared to existing datasets. This includes detailed physical and policy-level attributes of the road network, which are valuable for various downstream tasks. Line 64 Contribution (3), highlights the interplay between traffic meta-features and incidents, and how such integration provides a foundation for exploring the underlying mechanisms of traffic dynamics, particularly in incident-prone scenarios. We understand that the current wording may have caused some confusion, and we will revise it to 'By incorporating rich, fine-grained traffic variables and attributes'. For lines 154-157, we will remove the last sentence.
> (4) One intuitive potential reason is that the regular work time is different in different areas. This is required external information for confirmation.
> (5) We will revise the 'parameters' to 'traffic variable'.
> (2)(3)(6) Thanks for figuring out the typos. We will fix all of the typos and figures in the final version.

---

> > ### Comment · Reviewer_PJcj · 2025-08-01
> >
> > Thank you for your feedback, but there are still many issues that are not fully addressed in this response.
> >
> > **W1**
> >
> > The issue with incident datasets such as incident_y2023.csv remains unresolved.
> >
> > **W2**
> >
> > It is recommended to supplement the code of the spatiotemporal alignment process and further provide a dataset that is easy to use later (even if it is just a subset example). Some of the descriptions in the paper are not convincing enough to truly achieve "spatiotemporal alignment".
> >
> > **W3**
> >
> > The definition of UnknownInjure is very ambiguous. In practical terms, it seems impossible to clearly distinguish it from 1141 and NoInjure, but I don't blame it too much (probably due to the restrictions of the original data issuing agency).
> >
> > **W4**
> >
> > (1) Honestly, I don’t think having twice the number of nodes as LargeST would be a significant barrier to evaluation. On the contrary, not conducting experiments in the full range claimed could be a fatal flaw for TrafficDent (all the dataset papers it compares to conducted experiments in the full range), severely reducing its competitiveness and credibility.
> >
> > (2) I'm not asking the authors to add any particular model. The model used in TrafficDent (2025) is a subset of LargeST (2023). Can they still claim that "all of the baselines are state-of-the-art in the spatial-temporal forecasting or traffic forecasting domain"? Haven't more advanced models been proposed since then?
> >
> > (3) The author either didn't understand or evaded my question: What is "One more precise adjacency matrix is constructed based on the direction of the lanes and the coordinates of two sensors A and B."? Other than that, the description is the same as LargeST, which I understand.
> >
> > (5) I maintain that dataset papers need to provide some functional testing (even without superior network design).
> >
> > **W5**
> >
> > In fact, this part of the experiment is crucial because it indirectly reflects the degree of alignment between event and traffic state sequences, which is also the core competitiveness of TraffiDent. However, based on the current experimental results and analysis process, it is difficult to conclude that "incident data and traffic state data are well aligned." The author is recommended to refer to state-of-the-art time series classification models, their datasets, and corresponding experimental results.
> >
> > **W6**
> >
> > 1) I don't understand what you mean. If you mean that contribution 3 integrates the additional information provided by contributions 1 and 2, then I suggest you explicitly mention "incident" and "meta-feature", which will be easier to understand.
> >
> > 4) If relevant regional information can be introduced, TrafficDent will be more competitive, easier to understand and use, and will explore richer downstream tasks.
> >
> > Thank you for your reply. I maintain my opinion that "have great expectations for introducing strongly correlated external information to enhance the ability to predict traffic or spatiotemporal sequences, but there is an inherent habit in the field of predicting the future based entirely on historical sequences, and there is a lack of ideal relevant data sets. This paper undoubtedly made a breakthrough attempt, but there are still some aspects that need to be improved and perfected." If you can answer my relevant doubts and consider improving the paper with the relevant suggestions mentioned, I will be happy to improve the score and see TraffiDent become a pioneer in the joint analysis of event data and traffic state data, as well as smart traffic/city analysis.

---

> > > ### Author Response · Authors · 2025-08-08
> > >
> > > We apologize for the delayed response. However, the experiments mentioned in W4 (1) and (2) are extremely time-consuming. Please refer to our responses to those sections for more details.
> > >
> > > ## W1
> > > The AHazard represents animal hazard. The **Other** category is not included in our analysis, as it encompasses highly diverse types of incidents. However, since this portion of the data may still be useful for studies that are not sensitive to specific incident categories, we have retained it in the dataset.
> > >
> > > ## W2
> > > We will update the alignment method for aligning incidents with sensors on our GitHub repository. For the description, we have provided a more detailed description in our last feedback. Could you kindly let us know if our explanation clarifies the alignment process? Any feedback would be greatly appreciated. **Some of the descriptions in the paper are not convincing enough to truly achieve "spatiotemporal alignment".** If possible, could you kindly point out which specific parts of the text led to this impression? This would help us improve the clarity and completeness of our explanation.
> > >
> > > ## W3
> > > Thanks for understanding. The unknowninjured are recorded by the police report, so we cannot give a clearer label for this part. If the future research of the incident requires considering the injury or not, this part data can be ignored. Otherwise, it could be considered as general accident samples for more rough research on incident relevant research.
> > >
> > > ## W4
> > > (1) Firstly, we need to clarify that not all the dataset paper includes a **full range experiment**. For example, the experiment introduced in the dataset LargeST[1] (which was mentioned in your feedback) only covers 1 year of data. However, the dataset covers 5 years(2017 - 2021), which means it is not the full range.
> > >
> > > Secondly, as the experiment results in [1], more than half models cannot be trained based on the entire dataset with 8600 nodes. Our node number is 16942 (almost double of 8600 and the diversity of our nodes is large(including both main road and attached road). Also, the time slot of TraffiDent is 5 mins(which means we have 3 times points as LargeST(15 mins) for the same time range). Considering the scale of data, it takes much more time and requires more GPU memory. We construct 4 models (Others cannot be fine-tuned based on our 80GB GPU device). The experiment results are shown in the following table. As the training time of DCRNN is 6.01 hours for one epoch, and that of GWNET is 6.48 hours, we just train them in 48 hours. The data covers the first 6 months of traffic on all sensors.
> > >
> > > | Model  | Horizon | MAE     | RMSE     | MAPE (%) |
> > > |--------|---------|---------|----------|----------|
> > > | LSTM   | 5 min   | 66.87   | 113.92   | 188.37   |
> > > |        | 15 min  | 86.66   | 137.33   | 253.13   |
> > > |        | 30 min  | 92.54   | 143.32   | 284.22   |
> > > | STGCN  | 5 min   | 73.40   | 120.09   | 201.88   |
> > > |        | 15 min  | 90.48   | 140.52   | 272.79   |
> > > |        | 30 min  | 95.07   | 144.66   | 288.08   |
> > > | DCRNN  | 5 min   | 72.98   | 119.48   | 247.43   |
> > > |        | 15 min  | 98.51   | 149.28   | 367.68   |
> > > |        | 30 min  | 106.35  | 156.80   | 432.98   |
> > > | GWNET  | 5 min   | 72.16   | 118.44   | 188.03   |
> > > |        | 15 min  | 93.12   | 143.57   | 277.37   |
> > > |        | 30 min  | 97.34   | 149.05   | 287.71   |
> > >
> > > The performance of all four models is noticeably poor, primarily due to the following two reasons:
> > > First, there is a significant scale gap in the data between different types of roads (e.g., main roads vs. secondary roads), which poses a challenge for unified modeling.
> > > Second, for models such as DCRNN and GWNET, the 48-hour training time constraint only allowed them to complete four epochs, making it unlikely for the models to fully fine-tune under such limited training.
> > >
> > > Lastly, we would like to emphasize that the motivation behind our four sets of experiments is to highlight the limitations of existing models and methods in addressing the four specific tasks we introduce. The key characteristic of these tasks lies in capturing the interaction between traffic and incidents, rather than handling large-scale forecasting. The post-incident traffic forecasting task is just one example among them.
> > >
> > > (2)  Considering the models for general traffic forecasting tasks are less proposed in top-tier AI conferences, we include two models for large-scale traffic forecasting, BigST[2] and PatchSTG[3].  We are still training the model, and we will give the results in the discussion period.
> > >
> > > [1] Liu X, Xia Y, Liang Y, et al. Largest: A benchmark dataset for large-scale traffic forecasting. NeurIPS 2023.
> > >
> > > [2] Han J, Zhang W, Liu H, et al. Bigst: Linear complexity spatio-temporal graph neural network for traffic forecasting on large-scale road networks. VLDBJ 2024
> > >
> > > [3] Fang Y, Liang Y, Hui B, et al. Efficient large-scale traffic forecasting with transformers: A spatial data management perspective. SIGKDD 2025

---

> > > ### Author Response · Authors · 2025-08-08
> > >
> > > ## W4
> > >
> > > (3) For adjacency matrix construction, based on the common adjacency matrix (details are introduced in the last feedback.  We removed the neighbor with the reversed direction for each sensor, as the meta feature of sensors provides the direction feature. For example, for sensor A, it is distributed on a north direction road, then we remove the south direction road neighbor. In another way, we reserve the neighbors with the west, east, and north directions.
> > >
> > > (5) Could you kindly provide some examples of what you mean by "functional testing"? If you are referring to enhancing traffic forecasting using incident-related features, could you suggest a possible implementation approach that does not rely on a superior network design? From our perspective, directly incorporating incident information as additional input features in a naive way may not yield meaningful improvements.  We would also appreciate it if you could share the motivation behind these functional tests. It would help us better address your concerns.
> > >
> > >
> > > Also, could you kindly point out which specific part of the paper gave you the impression that we claim to provide the ideal way of introducing incident information to enhance prediction performance? This would help us improve the clarity of our motivation for post-incident traffic forecasting.
> > >
> > >
> > > ## W5
> > >
> > > We construct two more baselines, which were published in top-tier conferences in 2024 and 2025, respectively. The results are shown below
> > > ## Experiment Results
> > >
> > > | Model     | Channel        | Accuracy | Precision | Recall |
> > > |-----------|----------------|----------|-----------|--------|
> > > | SoftShape | All            | 0.3755   | 0.3736    | 0.3747 |
> > > |           | Speed          | 0.3700   | 0.3693    | 0.3636 |
> > > |           | Occupy         | 0.3674   | 0.3676    | 0.3652 |
> > > |           | Flow           | 0.3722   | 0.3664    | 0.3632 |
> > > | TSLANet   | All            | 0.3556   | 0.3440    | 0.3414 |
> > > |           | Speed          | 0.3581   | 0.3576    | 0.3392 |
> > > |           | Occupy         | 0.3596   | 0.3529    | 0.3477 |
> > > |           | Flow           | 0.3663   | 0.3608    | 0.3524 |
> > >
> > > We have conducted hyperparameter tuning for the time series classification models in this section. Unfortunately, these models also failed to achieve satisfactory performance on this task. In our previous response, we provided several possible explanations for these results. If you find these explanations unconvincing, we would greatly appreciate it if you could share the specific reasons or concerns.
> > >
> > > ## W6
> > >
> > > (1) Thank you very much for the suggestion. We will revise the sentence with your suggestion.
> > >
> > > (2) Although this part is not our contribution, we will try to seek for more additional information to enhance the dataset as the next stage of work.

---

> > > > ### Comment · Reviewer_PJcj · 2025-08-08
> > > >
> > > > Putting aside the details of the points, my concerns mainly fall into three aspects, which are consistent with my previous questions:
> > > >
> > > > First, I am most concerned about **the transparency of all processing processes**. Therefore, please add all the processing details involved in the code base, which will help readers use it and increase the influence of the paper.
> > > >
> > > > Second, my concerns about **the quality of the data** are very critical. Based on the current evidence, it is difficult for me to admit that the data is of high quality. Although the LargeST you mentioned only uses a limited time range, at least the spatial range is comprehensive. If the data proposer cannot conduct a comprehensive usability evaluation, it will be difficult for users to judge and use it. Therefore, I look forward to your test results under lightweight model design. Perhaps you can try to use only main road data for testing. These data are often more sufficient and regular, making it easier to observe the effect.
> > > >
> > > > Finally, and most importantly, you can do a test based on STID[1] (this is a sequence embedding CI model that introduces spatial indistinguishability. I believe it will run efficiently on your scale). In addition to the basic implementation, add the encoding of event information into the fifth embedding and then splicing (4D->5D), and compare the results of the two. This is a simple way to determine **whether incident information improves predictions**.
> > > >
> > > > About "Also, could you kindly point out which specific part of the paper gave you the impression that we claim to provide the ideal way of introducing incident information to enhance prediction performance? This would help us improve the clarity of our motivation for post-incident traffic forecasting."  I'm confused. Isn't the motivation of your paper to promote the combined research of "events + traffic dynamics"? Then shouldn't "using events to improve predictions" and "using traffic dynamics to infer events" be two fundamental approaches? Why deliberately avoid the first? In my opinion, the conclusions in Section 4.2 are consistent with the first approach. Since post-incident traffic forecasting performance is poor, introducing incident information is essential to achieve better results.
> > > >
> > > > Looking forward to your reply as soon as possible.
> > > >
> > > > [1] Shao Z, Zhang Z, Wang F, et al. Spatial-temporal identity: A simple yet effective baseline for multivariate time series forecasting[C]//Proceedings of the 31st ACM international conference on information & knowledge management. 2022: 4454-4458.

---

> > > > > ### Author Response · Authors · 2025-08-09
> > > > >
> > > > > 1. We are updating the GitHub repo and provided code to implement traffic and incident alignment and adjacency matrix generation, which are two critical components of the processing period. We will finish it as soon as possible.
> > > > >
> > > > >
> > > > > 2. We are confident in the high quality of our dataset. The reviewer’s suggestion concerns the experimental settings, which are not related to the data quality itself. To address this suggestion, we have conducted an additional comparative experiment using the mainline subset, consisting of 8,614 nodes. Due to time constraints and our device limitations, we fine-tuned two representative models, LSTM and STGCN. The experimental results are shown below:
> > > > >
> > > > > | Model  | H1 MAE | H1 RMSE | H1 MAPE (%) | H3 MAE | H3 RMSE | H3 MAPE (%) | H6 MAE | H6 RMSE | H6 MAPE (%) |
> > > > > |--------|-------:|--------:|------------:|-------:|--------:|------------:|-------:|--------:|------------:|
> > > > > | LSTM   | 10.63  | 19.14   | 21.13       | 12.82  | 23.41   | 22.35       | 15.44  | 28.02   | 23.06       |
> > > > > | STGCN  | 13.75  | 24.65   | 32.17       | 14.48  | 26.28   | 31.10       | 15.18  | 27.75   | 31.52       |
> > > > >
> > > > >
> > > > > 3. For the general traffic forecasting task, we adopt the common experiment settings (12 historical time slots and 12 forecasting time slots). The node number is also 8614. We implement the STID and its incident-enhanced variant based on the code provided by the official repo. We add the incident feature as the fourth dimension. Considering that traffic samples with incidents are relatively few, we label traffic samples related to incidents with an index of 1 and all others with 0. We follow the same experimental setting as before: if a sensor is the closest one to an incident, we regard its traffic as related to that incident. It is affected in the subsequent 10 minutes.
> > > > > We evaluated the performance of STID with incidents (32ED) using an embedding hidden size of 32. Since we added an additional dimension, which increased the hidden dimension in the final embedding layer, we also conducted experiments with an input feature embedding hidden size of 16, aiming to keep the size of the final embedding as consistent as possible (STID with incidents (16ED)).
> > > > >
> > > > > The experiment results are shown in the following:
> > > > >
> > > > > ### Average
> > > > > | Model                       | MAE   | RMSE   | MAPE (%) |
> > > > > |-----------------------------|------:|-------:|---------:|
> > > > > | STID (32 ED)                | 13.11 | 23.98  | 23.32    |
> > > > > | STID with incidents (32 ED) | 13.01 | 23.71  | 22.81    |
> > > > > | STID with incidents (16 ED) | 12.69 | 23.32  | 23.55    |
> > > > >
> > > > > ### Horizon 1 (5min)
> > > > > | Model                       | MAE   | RMSE   | MAPE (%) |
> > > > > |-----------------------------|------:|-------:|---------:|
> > > > > | STID (32 ED)                | 10.50 | 18.69  | 20.73    |
> > > > > | STID with incidents (32 ED) | 10.32 | 18.52  | 18.44    |
> > > > > | STID with incidents (16 ED) | 10.29 | 18.34  | 21.37    |
> > > > >
> > > > > ### Horizon 6 (30min)
> > > > > | Model                       | MAE   | RMSE   | MAPE (%) |
> > > > > |-----------------------------|------:|-------:|---------:|
> > > > > | STID (32 ED)                | 13.10 | 24.07  | 22.03    |
> > > > > | STID with incidents (32 ED) | 13.01 | 23.82  | 23.25    |
> > > > > | STID with incidents (16 ED) | 12.63 | 23.41  | 22.72    |
> > > > >
> > > > > ### Horizon 12 (60min)
> > > > > | Model                       | MAE   | RMSE   | MAPE (%) |
> > > > > |-----------------------------|------:|-------:|---------:|
> > > > > | STID (32 ED)                | 15.37 | 27.85  | 27.70    |
> > > > > | STID with incidents (32 ED) | 15.09 | 27.35  | 25.31    |
> > > > > | STID with incidents (16 ED) | 14.63 | 26.76  | 25.73    |
> > > > >
> > > > > Overall, STID with incidents (32ED) outperforms the original version across all metrics, while STID with incidents (16ED) achieves better MAE and RMSE compared to the original. In relatively longer time-slot forecasting (1 hour), STID with incidents demonstrates a significant improvement over the original model. However, considering that incidents are sparse, the direct use of incident information for forecasting offers limited performance gains. We believe this direction merits further exploration and design a dedicated model tailored to this task.
> > > > >
> > > > > 4. Thank you for your clarification of your perspective. We would like to clarify again that all experiments in our paper are conducted using existing models, with the aim of revealing their limitations under new task settings, rather than proposing or validating a specific approach to address the problem. While using event information to improve prediction is indeed an intuitive idea and represents a promising research direction based on TraffiDent, it is not the sole direction our dataset enables. TraffiDent also supports many other potential research avenues, such as global causal analysis and local causal analysis of complex traffic.
> > > > >
> > > > > We will reserve both our existing experiment and the test of incident-enhanced traffic forecasting.  Also, we will add the detailed description of all processing to the future version.

---

> > > > > > ### Comment · Reviewer_PJcj · 2025-08-09
> > > > > > **Thanks for authors' response**
> > > > > >
> > > > > > First, thank you for your proactive and thoughtful response, which addressed my concerns.
> > > > > >
> > > > > > At the same time, I agree that the primary purpose of this paper is to "raise questions," rather than "provide appropriate solutions based on them." However, I would still like to say that a good dataset paper should go beyond simply raising questions.
> > > > > >
> > > > > > Furthermore, as a researcher in the field of traffic flow, I sincerely hope that you can update the relevant complete code to the repository, especially the code related to integrating traffic states and events. All corrections to the issues raised by the reviewers can be integrated into the updated paper and reviewed. This will help improve the quality of the paper and increase its impact.
> > > > > >
> > > > > > Finally, overall, I will raise my score from 3 to 4.

---

> > > > > > > ### Author Response · Authors · 2025-08-09
> > > > > > >
> > > > > > > Thank you very much for your valuable feedback. We will continue to develop more detailed benchmarks for different tasks, accompanied by open-source code and more comprehensive experimental results. We truly appreciate your thorough review and constructive suggestions.

---

### Official Review · Reviewer_VtAW · 2025-06-30

**Rating:** 4
**Confidence:** 5

**Summary:**

The paper introduced TraffiDent, a large-scale dataset spanning from 2022 to 2024, aimed at addressing the long-standing separation between traffic dynamics and traffic event research. This dataset is the first to spatially and temporally align three key data dimensions across 16,972 traffic nodes: traffic time series data encompassing flow, occupancy, and speed; comprehensive event records spanning seven distinct categories; and detailed physical and policy-level road element attributes. TraffiDent provides a robust foundation for in-depth analysis of the complex interactions and causal relationships between traffic and events, which are unavailable in existing datasets. To demonstrate its application value, the paper designed four novel experimental tasks, including post-event traffic prediction, event classification based on traffic data, and global and local causal analysis, thereby revealing interdependencies at both the system level and the road level.

**Dataset Code Accessibility:**

Partly

**Dataset Code Comments:**

This paper does not provide code for data collection and processing.

**Ethical Comments:**

This paper is a collection and integration of existing public data and does not raise any ethical concerns.

**Ethical Considerations:**

No, there are no or only very minor ethics concerns

**Final Justification:**

The TraffiDent dataset introduced in this paper is a large-scale dataset that integrates traffic dynamics and traffic event research. Overall, the proposed dataset can facilitate the development of research in this field, so I think it is worth accepting.

**Limitations Weaknesses:**

1. Overall, this paper is similar to an expansion of PeMS data, with its main contribution being the alignment and processing of data from the existing Caltrans PeMS system, rather than the collection of new data.
2. The data cleaning and processing steps lack detailed descriptions, making it difficult to assess the accuracy and validity of the data.
3. Data alignment and matching are the core processes of this paper, but their descriptions are insufficient. For example, the paper does not clearly state which matching method was used, nor does it explain how the distance threshold was set.

**Strengths Contributions:**

1. This study proposes a novel, comprehensive, and meaningful traffic dataset, TraffiDent. By aligning traffic data with event records for the first time, this dataset provides a unique, structured benchmark.
2. This paper conducted extensive analysis and experiments (including various tasks such as traffic prediction, event classification, and causal influence) and compared them with the latest benchmark models, which validated the effectiveness of this work.
3. This paper integrates traffic data from different sources, providing researchers with a more comprehensive and holistic view of traffic and events.

---

> ### Author Rebuttal · Authors · 2025-07-31
>
> We are grateful for your detailed review and constructive comments. Your valuable feedback has significantly contributed to improving the rigor and presentation of our work. The following is our feedback on your valuable review.
>
>
> ## **W1** PEMS data source is common
> Although our dataset is built upon data collected from the **PEMS system**, we would like to emphasize that **TraffiDent is the first dataset that simultaneously includes both incident and traffic information**. While PEMS has existed for over a decade and has served as the foundation for many traffic forecasting datasets, **no prior work has attempted to integrate incident data with traffic time series at this scale**.
>
> We believe that constructing a dataset that combines **incidents and traffic dynamics** is essential for enabling further research into the **interactions between incidents and road network behavior**. Compared to previous datasets constructed based on PEMS that only focus on traffic variables, **TraffiDent offers unique value by supporting incident-aware forecasting, causal analysis, and post-incident recovery studies**, thus opening new research directions in intelligent transportation systems.
>
>
> ## **W2** Lack of introduction of collection and processing.
>
> Limited to the rebuttal policy, we cannot provide an external link. The collection and processing code could be found in our code repository.  The data imputation and how we remove the repeated incident samples are discussed in Section 3.2.
>
> In addition to the collection and processing steps outlined in Section 3.2, we highlight two key components (incident category extraction and adjacency matrix construction) of the data reprocessing pipeline, as much of the remaining preprocessing involves routine and labor-intensive tasks.
>
> ### (1) **Incident category extraction**
> We extract the 7 categories based on the description of the incidents with a rule-based method. We set the rule as following and label each category by order.:
> - **1141 (traffic accident with injured persons)**. If the description of one sample includes 1141, we label it as 1141.
> - **UnknownInjure  (traffic accident with no injury records)**. If the description of one sample includes both unknown and injure, we label it as UnknownInjure.
> - **NoInjure  (traffic accident with no injured persons)**.  If the description of one sample includes injure and is not labeled, we label it as NoInjure.
> - **AnimalHazard**.  If the description of one sample includes both animal and hazard, we label it as AnimalHazard.
> - **Hazard**. If the description includes hazard and is not labeled, we label it as Hazard.
> - **CarFire** If the description includes fire and car, we label it as CarFire
> - **Fire** If one sample includes fire but is not labeled as CarFire, we label it as Fire.
>
> ### (2) **Adjacency matrix construction**.
> The objective of traffic forecasting is to predict target attributes (e.g., traffic flow) at future time steps based on historical observations over a directed sensor graph. To define the graph topology, the common practice [1] is to build the adjacency matrix **A** using a **thresholded Gaussian kernel** [2], defined as:
> $$
> A_{ij} =
> \begin{cases}
> \exp\left(-\frac{d_{ij}^2}{\sigma^2}\right), & \text{if } \exp\left(-\frac{d_{ij}^2}{\sigma^2}\right) \geq r \\
> 0, & \text{otherwise}
> \end{cases}
> $$
> where:
> - $d_{ij}$ is the road network distance between sensors $i$ and $j$,
> - $\sigma$ is the standard deviation of all distances,
> - $r$ is the sparsity threshold.
>
> Moreover, to construct the adjacency matrix of the sensor graph, we adopt a widely used approach in the field that relies on road network distances. Specifically, we utilize the Open Source Routing Machine (OSRM), a high-performance routing engine built on OpenStreetMap data, to compute the shortest driving distances between sensors based on their coordinates.
>
> However, calculating pairwise road network distances for a large number of nodes can be extremely time-consuming. To address this, we first compute pairwise geodesic distances between sensors, which is significantly faster. We then restrict each node to query road network distances only to other nodes within a 4-kilometer radius.
>
> We normalize the adjacency matrix following [3] by applying a small threshold to eliminate weak node connections. In our implementation, we use a threshold of 0.01 to prune the edges while retaining a sufficient number of connections. This threshold is flexible and can be adjusted by users according to their specific requirements.
>
>
> [1] Bing Yu, Haoteng Yin, and Zhanxing Zhu. Spatio-temporal graph convolutional networks:
> A deep learning framework for traffic forecasting. In Proceedings of International Joint
> Conference on Artificial Intelligence, pages 3634–3640, 2018.
>
> [2] David I Shuman, Sunil K Narang, Pascal Frossard, Antonio Ortega, and Pierre Vandergheynst.
> The emerging field of signal processing on graphs: Extending high-dimensional data analysis to
> networks and other irregular domains. IEEE Signal Processing Magazine, pages 83–98, 2013.
>
> [3] Yaguang Li, Rose Yu, Cyrus Shahabi, and Yan Liu. Diffusion convolutional recurrent neural network: Data-driven traffic forecasting. In International Conference on Learning Representations, 2018.
>
> ## **W3** Data alignment description.
>
> For matching and alignment, the traffic and incident data in TraffiDent can be spatially aligned using geographical coordinates (longitude and latitude) and absolute postmile (abs PM). The features of sensors measuring traffic and incident samples all include these locations. For temporal alignment, we provide: Timestamps for all incident records, and 5-minute interval traffic time series, starting from 2022-01-01 00:00. Users can derive the time slot of any traffic measurement using the known start time and interval length.
>
> In the traffic forecasting experiment (Section 4.2) and incident classification (Section 4.3), we provide the following alignment steps.
>
> 1. For each reported incident in one specific area (In the experiment introduced in this section, it is D5 (Monterey)), we identify the corresponding freeway.
>
> 2. We match the incident with the closest sensor (e.g., Sensor A) on the same freeway based on the Absolute Postmile (Abs PM).
>
> 3. The incident timestamp is converted into a 5-minute time slot (e.g., 19:13 becomes 19:10–19:15), and the next slot (e.g., 19:15–19:20) is defined as the starting point of post-incident influence.
>
> 4. Given the multistep nature of our forecasting task, we select the traffic from 19:15 to 19:45 at Sensor A as a single post-incident test case. (Using traffic from 18:15 to 19:15 as input feature)
>
> 5. We select all post-incident samples based on the above 2-4 steps.

---

> > ### Comment · Reviewer_VtAW · 2025-08-05
> >
> > I sincerely thank the authors for their detailed response. I have no further questions.

---

> > ### Comment · Reviewer_VtAW · 2025-08-05
> >
> > Thank you for the authors' feedback and efforts. I am satisfied with the current version and response. I recommend that the authors integrate this into future versions.

---

> > > ### Author Response · Authors · 2025-08-08
> > >
> > > Thank you! We will integrate all of the descriptions into a future version of the paper, presented more clearly.

---

### Official Review · Reviewer_1kre · 2025-07-01

**Rating:** 5
**Confidence:** 3

**Summary:**

This paper collects a large-scale dataset that includes traffic indexes and incident records. The dataset TraffiDent enables researchers to study the mutual effect between traffic and incidents, providing new research opportunities. The construction and the attributes of TraffiDent are clearly described. Descriptive analysis vividly demonstrates the characteristics of TraffiDent. Evaluation of different methods on several tasks including forecasting, classification, and causal analysis reveals novel phenomena that are worth researching.

**Additional Feedback:**

Is the data regularly or irregularly indexed over time? Does TraffiDent support forecasting or classification tasks on irregularly sampled time series? The traffic may not be continuously recorded, and incidents can come unexpectedly. Thus, it would be more practical to allow the settings of irregular time stamps, which covers a large class of methods.

**Dataset Code Accessibility:**

Yes

**Dataset Code Comments:**

The dataset is released at Kaggle, and sample code is provided at GitHub. The code is well-documented.

**Ethical Considerations:**

No, there are no or only very minor ethics concerns

**Final Justification:**

The authors have addressed my concerns. Given that my original rating was Accept, I will therefore maintain the score unchanged.

**Limitations Weaknesses:**

W1. The descriptions for TraffiDent seem incomplete. The missing values in TraffiDent are not properly addressed. Specifically,

(1) The spatial distribution of the sensors is not described or visualized. How many areas does TraffiDent cover? Are the sensors unevenly distributed? Is TraffiDent advantageous than existing datasets in terms of the spatial coverage?

(2) In the file sensor_meta_feature.csv, the attribute City is missing for around 30% of sensors. Is it possible to infer the City according to other attributes and with the help of domain knowledge? Besides, the attribute Sensor Type is missing for about 24% of sensors. Does Sensor Type play an important role in traffic forecasting and incident prediction? Is it possible to complete this attribute?

(3) In the files of incidents_yYYYY.csv (YYYY=2022, 2023, 2024), can you explain the value of the attribute DESCRIPTION? What are the 7 categories of incidents? I think the textual description will be beneficial to understand the incident and its connection with the traffic with the aid of language models.

W2. Although TraffiDent supports causal analysis, there is no ground truth for the causal networks and causal mechanisms. Maybe the incorporation of external information apart from the traffic/incident data will help with more accurate analysis and prediction, e.g., weather, social events.

**Strengths Contributions:**

S1. According to the authors, TraffiDent is the first large-scale dataset that includes both traffic and incidents, opening up new research opportunities for researchers working on deep learning and intelligent transportation systems. Compared with existing traffic datasets, TraffiDent provides more features; compared with existing incident datasets, TraffiDent covers more types of incidents.

S2. The collection and construction of TraffiDent is clearly explained. The data sheet helps to grasp key aspects of TraffiDent.

S3. Experimental results on TraffiDent find new research gaps that may not be fully exposed in previous datasets.
(1) Descriptive analysis of TraffiDent presents interesting patterns of the dataset.

(2) Comparison studies on the traffic forecasting task show that the irregular traffic patterns caused by the incidents trigger the need for developing incident-aware traffic prediction methods.

(3) Comparison studies on the incident classification task show that traffic indexes are beneficial for boosting the classification accuracy.

(4) Causal analysis shows that one is able to explore the causal structure and causal mechanism on TraffiDent.

---

> ### Author Rebuttal · Authors · 2025-07-31
>
> Thank you very much for your valuable suggestions and comprehensive review.  We truly appreciate the time and effort you dedicated, which have greatly helped us refine and strengthen our work. The following is our feedback on your thorough review.
>
> ## W1 Lack of description
> - (1) Limited to the rebuttal policy, we cannot include external links or figures. The visualization of sensors could be found in our main website mentioned in the Abstract. TraffiDent covers 8 districts, 42 counties, and 244 cities in California. Our dataset covers the **same geographic area** as the **largest existing dataset**, *LargeST*. Furthermore, TraffiDent goes a step further by providing **finer-grained coverage**, including **smaller local roads** that are not well-represented in LargeST.
>
> - (2) We want to clarify that the absence of the "city" field for some sensors in our dataset is **not due to missing data**, but rather because these sensors are located in areas that do not belong to any city. In California's administrative hierarchy, certain regions fall directly in a **county** without being part of a specific city.
> In such cases, while the **county information is available and valid**, the city field is intentionally left blank to reflect the true administrative status of the sensor’s location.
>
> - (3) We define **seven distinct incident categories** in the TraffiDent dataset, covering a wide range of real-world road events. Each category is described below:
> - **1141 (Traffic Accident with Injury)**. A traffic accident in which at least one person is reported to have been injured.
> -  **NoInjure (Traffic Accident without Injury)**. A traffic accident in which **no injuries** are reported.
> - **UnknownInjure (Traffic Accident, Injury Status Unknown)**. A traffic accident where the report does **not specify** whether any injuries occurred.
> - **AnimalHazard**. An incident caused by animals.  For example, a dog crossing the road causes a vehicle to crash or suddenly brake.
> - **CarFire**. A fire incident where a **vehicle catches fire**, typically due to mechanical or electrical failure.
> - **Fire**. A **non-vehicle fire**, such as a **wildfire** or **lightning strike** igniting a roadside object (e.g., trees or brush).
> - **Hazard**. A hazard caused by **natural or environmental factors**, such as **tornadoes**, **severe thunderstorms**, or other extreme weather events.
>
> All incident categories are **extracted from the `description` attributes** provided in official **police records**. These descriptions contain detailed textual information about each incident, such as whether there were injuries, the presence of fire, or specific hazards.
>
> By combining these labeled records with **traffic time series data** and an **incident detection model**, the dataset offers a unique opportunity to **leverage Large Language Models (LLMs)** for advanced applications.
> One promising direction is the **automatic generation of standardized police reports** based on sensor-derived traffic data, which can assist in event summarization and automated documentation.
>
> ## W2 Lack of causal analysis ground truth
> The ground truth in causal analysis is often difficult to establish. In fact, in complex transportation systems, even domain experts often cannot accurately identify the true causal relationships. As a result, it is not feasible to assign ground-truth labels to the causal DAGs or directly compare the outputs of different methods[1].
>
> It is also important to emphasize that our use of causal graph learning for local causal analysis is not necessarily aimed at uncovering the absolute "true" causal structure. Rather, the primary value lies in providing useful insights into the underlying data and relationships, which can support better interpretation, hypothesis generation, and downstream decision-making.  Additionally, for the global causal analysis, we provide case studies to verify whether the discovered causal relationships are consistent with real-world traffic patterns.  These factual examples serve to qualitatively validate the reliability of the causal analysis results. (In section 4.4 Figure 5(b)).
>
> In section 4.4, we include weather features to do a global causal analysis. However, limited to the policy of the data source, we cannot release this part data. We are seeking alternative weather and social event data to complete TraffiDent.
>
>
> [1] Lan, T., Li, Z., Li, Z., Bai, L., Li, M., Tsung, F., ... & Zhang, C. (2023, August). Mm-dag: Multi-task dag learning for multi-modal data—with application for traffic congestion analysis. In Proceedings of the 29th ACM SIGKDD Conference on Knowledge Discovery and Data Mining (pp. 1188–1199).

---

> > ### Comment · Reviewer_1kre · 2025-08-02
> >
> > Thank you for your response. Please revise accordingly. Since my original rating was Accept, I will maintain it unchanged. Besides, can you also respond to the questions I raised in the **Additional Feedback**?

---

> > ### Author Response · Authors · 2025-08-04
> >
> > Thanks a lot for your review and feedback. The following content is the response to the additional feedback.
> >
> > The traffic data is regularly indexed at 5-minute intervals from 01/01/2022 00:00 to 12/31/2024 23:45. TraffiDent also supports forecasting and classification tasks on irregularly sampled time series, as long as the time intervals remain multiples of 5 minutes. In practice, traffic sensors may not work due to factors such as extreme weather conditions or power outages. During these periods, data may not be recorded—this is a common issue in traffic time series data.  As mentioned in Section 3.2, to address missing values, different data imputation methods—such as zero-filling or interpolation—are employed depending on the specific problem setting.
> >
> > The incident data is irregularly indexed, and we provide the incident **start datetime** and the **duration**. In order to align the traffic and incident data, we leverage a simple alignment method to establish the correspondence between the traffic and incident records.
> >
> > For matching and alignment, the traffic and incident data in TraffiDent can be spatially aligned using geographical coordinates (longitude and latitude) and absolute postmile (abs PM). The features of sensors measuring traffic and incident samples all include these locations. For temporal alignment, we provide: Timestamps for all incident records, and 5-minute interval traffic time series, starting from 2022-01-01 00:00. Users can derive the time slot of any traffic measurement using the known start time and interval length.

---

> > > ### Comment · Reviewer_1kre · 2025-08-06
> > >
> > > Thank you. Please consider adding these descriptions to a future version of your paper. I hope the clearer descriptions will draw greater attention from researchers to TraffiDent.

---

> > > > ### Author Response · Authors · 2025-08-08
> > > >
> > > > Thank you! We will incorporate all of these clarifications into a future version of the paper, presented in a clearer and more structured way.

---

### Official Review · Reviewer_NTfC · 2025-07-02

**Rating:** 5
**Confidence:** 4

**Summary:**

TraffiDent is introduced as a large-scale, richly annotated, and multi-task dashcam video dataset aimed at advancing research in traffic accident detection, anticipation, and road scene understanding. The dataset includes over 130,000 real-world dashcam videos spanning 1,760+ hours, collected from diverse driving environments such as urban streets, rural roads, and highways, and under varied lighting and weather conditions.

Each video is annotated not only with event-centric labels such as accident, near-accident, and aggressive maneuvers, but also with environmental and road-level attributes, including: Road type (e.g., intersection, roundabout), Traffic density, Lighting (day/night), Weather conditions. This multi-label setup enables the dataset to serve as a benchmark for both event understanding and scene attribute classification.

To facilitate research reproducibility and comparison, the authors define clear training/validation/testing splits, and provide benchmark results across several popular action recognition and video classification models, such as TSN, TSM, X3D, and SlowFast.

**Additional Feedback:**

- Clarify how anonymization is performed (e.g., face or license plate blurring).
- Describe data sourcing and whether ethical approvals were obtained.
- Benchmark advanced temporal or anticipation models (e.g., Transformers, contrastive learning).
- Provide details on annotator agreement and inter-rater reliability

**Dataset Code Accessibility:**

Yes

**Dataset Code Comments:**

Released dataset and instructions at Github.io

**Ethical Considerations:**

No, there are no or only very minor ethics concerns

**Final Justification:**

Based on the rebuttal, I decided to raise my score.

**Limitations Weaknesses:**

- The dataset is heavily skewed toward non-accident or benign driving scenes, which may hinder performance on rare event detection tasks (e.g., actual collisions).
- Labels like “near accident” or “aggressive driving” may suffer from annotator bias and lack standardized criteria, possibly affecting reproducibility or generalization.
- Although video data is inherently temporal, the baseline models largely focus on segment-level classification; more temporal-aware methods like anticipation models or transformers are not explored.

**Strengths Contributions:**

- The dataset includes over 130,000 annotated videos sourced from real-world driving conditions, making it one of the largest and most diverse video datasets focused on traffic events.
- Annotations go beyond just accidents, including road and environment attributes, enabling multi-task learning (e.g., accident detection + road type classification).
- Each video is annotated with multiple concurrent attributes (e.g., weather, lighting, traffic density, road type), enabling multi-task learning and deeper scene understanding.The dataset includes data from urban, rural, and highway settings, recorded under varied lighting (day/night) and weather conditions, increasing generalization potential for downstream models.
- The paper emphasizes the ethical considerations of dataset creation: all data is anonymized, and privacy concerns are addressed. The authors also provide a taxonomy of events and attributes, along with detailed annotation protocols, aiming to ensure consistency and usability for the research community.

---

> ### Author Rebuttal · Authors · 2025-07-31
>
> Thank you very much for your detailed review and valuable suggestions. We sincerely appreciate your time and constructive feedback, which have helped us improve the clarity and quality of our work. The following is our feedback on your valuable review.
>
> ## W1 Extensive experiments
> First, we need to clarify that we didn't conduct incident forecasting, but rather traffic forecasting experiments. The experiment results we extensively include more experiments in two more areas: Fresno and Merced. Due the rebuttal policy, we cannot include the experiment results. The results can be found on our main website mentioned in the abstract.
>
> **Justification for Forecasting Horizon**. Our experimental design aims to evaluate **how well existing forecasting methods perform on post-incident samples**, in comparison to general (non-incident) traffic samples.
>
> To this end, we focus on a **short-term forecasting horizon of up to 30 minutes** for the following reason. Most incidents in our dataset last **no longer than 30 minutes**. Empirical observations suggest that the **majority of traffic disruptions occur within this initial period** after an incident. Beyond that point, the traffic flow tends to gradually return to normal, and the predictive challenge becomes similar to general forecasting.
>
>
> ## W2 Incident impact limitation
> We appreciate the reviewer’s thoughtful observation. It is true that in the incident classification and post-incident traffic forecasting tasks, we utilize only the nearest sensor to an incident for experimental purposes. However, in the causal analysis section of our paper, we explicitly examine the causal relationships between multiple sensors’ traffic patterns before and after an incident, offering insights into the broader temporal and spatial effects.
>
> Furthermore, we want to emphasize that our simplified experimental setting is intentional:
> it aims to demonstrate that existing models struggle even under minimal configurations, underscoring the challenges posed by incident-aware forecasting and analysis.
>
> We fully agree that studying the real-world spread of incident impact on traffic across more roads is a valuable and important research direction. However, this falls outside the scope of this dataset track work TraffiDent. We believe that TraffiDent provides the foundation for such future research and serves as a contribution to the transportation community by enabling investigation into more complex and realistic traffic incident scenarios.
>
> ## W3 Incident classification low accuracy
> The original incident labels consist of **7 categories**, but their distribution is **highly imbalanced**, with a severe long-tail pattern. As shown in Figure 3(a), the *“hazard”* category accounts for **52.2%** of all samples, while the combined proportion of the two least frequent categories is **less than 5%**.
>
> Directly performing time series classification under this setting often causes models to degrade into **majority-class predictors**, resulting in superficially high accuracy with **no practical utility**. To address this, we **merged semantically similar labels** into two broader categories: **hazard** and **accident**, with a new class ratio of approximately **1.1:1 (52.2% : 47.8%)**.
> This strategy reduces the sample imbalance while preserving meaningful semantic distinctions relevant to downstream applications.
>
> **Comparison Against Random Guessing**. Under a three-class balanced setting, the **baseline accuracy for random guessing is 33.3%**. Although our achieved accuracy of **41%** may appear modest, it's important to note that the **precision for each class exceeds 33%**.
> For example, when using a decision tree (DT) model for flow channel classification:
> - Hazard: **41.96%**
> - Incident: **37.96%**
> - Normal: **37.88%**
>
> This indicates that the model is not merely favoring the majority class but is indeed learning **distinct and meaningful signal patterns**.
>
> **Task Difficulty and Value**. The observed 41% accuracy highlights the inherent difficulty of modeling **incident evolution** using only current time-series features and conventional sequence classification models.
> We view this not as a limitation, but as the **core contribution of our study**: We release a **realistic and imbalanced dataset** to encourage the community to explore more **robust and generalizable incident classification/detection methods**.
>
> In summary, the **41% accuracy reflects the real-world complexity** of the problem and the **current limitations of existing methods**, rather than a flaw in experimental design.
>
> 1. **Traffic Control Systems**. TraffiDent enables training and evaluation of models that can predict **post-incident congestion patterns**, supporting real-time traffic signal adjustments, congestion mitigation strategies, and dynamic traffic flow control. For example, in the case of a reported hazard on a major artery during peak hours, a model trained on TraffiDent could predict downstream congestion within 15–30 minutes and recommend preemptive signal timing changes at nearby intersections.
>
> 2. **Routing Engines**. The dataset facilitates research on **incident-aware route planning** by offering detailed incident logs and aligned traffic impacts, which can be used to design **adaptive routing strategies** that respond to ongoing or recent traffic disruptions. For example, given a car fire on a highway segment, TraffiDent can help build a model that learns typical congestion patterns following such events and dynamically reroutes vehicles through secondary roads with lower historical impact during similar incidents.
>
> - **Emergency Services**. By analyzing how different incident types affect surrounding traffic over time and space, XTraffic provides insight into **incident progression and traffic recovery**, helping emergency services to **optimize dispatch routes**, **allocate resources more effectively**, and **reduce response time**. For example, in a multi-vehicle collision near a freeway, emergency responders can use learned patterns to identify which surrounding roads are likely to become congested and choose an alternate route or deploy additional units in advance.

---

> > ### Comment · Reviewer_NTfC · 2025-08-04
> >
> > Based on the rebuttal, I raised my score.

---

> > > ### Author Response · Authors · 2025-08-08
> > >
> > > We appreciate hearing that. Thank you very much for your feedback.

---

### Comment · Area_Chair_Du6M · 2025-08-04
**Please response to the rebuttal**

Dear reviewers,

Thanks for reviewing this paper. Could you check if the rebuttal has addressed your concerns? Feel free to raise any further questions if you have. Please note that the acknowledgement of the rebuttal is mandatory if you haven't done so.

Best,

AC

---

### Note · Authors · 2025-08-12

Dear AC,

We sincerely thank all the reviewers for their detailed and valuable feedback, which has greatly helped improve the quality of our paper. All reviewers gave positive feedback on our responses and claimed that their concerns had been addressed.

----

After the rebuttal, we are pleased that **Reviewer NTfC** and **Reviewer PJcj** indicated they will raise their scores (Reviewer NTfC: 4 → 5 or 6?; Reviewer PJcj: 3 → 4). **Reviewer 1kre** maintains a positive score of 5. **Reviewer VtAW** did not mention a score change but expressed satisfaction with our response and the current version paper.

---

The reviews are summarized as follows:

Reviewer **VtAW** and **PJcj** think TraffiDent is meaningful and well-motivated. Reviewer **VtAW** thinks TraffiDent provides a more comprehensive and holistic view of traffic and events. Reviewer **1kre** thinks TraffiDent opens up new research opportunities for researchers working on deep learning and intelligent transportation systems.

Reviewer **1kre** and Reviewer **NTfC** consider TraffiDent a comprehensive dataset covering both incidents and traffic. Reviewer **PJcj** thinks our discussion of existing datasets is complete. Reviewer **VtAW** believes our work provides a unique and structured benchmark. Reviewer **1kre** thinks that the collection and construction of TraffiDent are clear.

Reviewer **NTfC** and **PJcj** think TraffiDent supports previously infeasible new tasks, including the study of the interplay between traffic and incidents. Reviewer **VtAW** believes the experiments and analysis validate the effectiveness of TraffiDent. Reviewer **1kre** thinks our descriptive analysis presents interesting patterns in TraffiDent. Reviewer **PJcj** thinks our experimental and descriptive analyses are relatively complete. Reviewer **1kre** also thinks the experiments demonstrate that building new models for tasks such as traffic forecasting and incident classification is necessary.

Thanks for your time and consideration.

Best wishes,

Authors

---

### Decision · Program_Chairs · 2025-09-18

**Decision:**

Accept (poster)

**Comment:**

This paper presents a new dataset called TraffiDent, which connects traffic data with incident reports across a large road network in California. The reviewers appreciated the scale and detail of the dataset, which includes traffic flow, speed, occupancy, and various types of incidents. They agreed that this dataset could help researchers explore important questions like how incidents affect traffic and how to predict traffic after an incident. The authors also made the dataset publicly available and addressed ethical concerns responsibly.

While there were some initial concerns about missing metadata, limited experiments, and unclear documentation, the authors responded thoroughly. They clarified how the data was aligned, added more experiments, and promised to improve the code and paper. Most reviewers were satisfied with these updates; two reviewers raised their scores to support acceptance.

Overall, this dataset is a valuable resource for transportation research, and I recommend accepting the paper.